# ChunkKV: Semantic-Preserving KV Cache Compression for Efficient Long-Context LLM Inference

**Xiang LIU**♡∗    **Zhenheng TANG**♣∗    **Peijie DONG**♡    **Zeyu LI**♡
**Yue LIU**◇†    **Bo LI**♠♣    **Xuming HU**♡†    **Xiaowen CHU**♡†

♡ The Hong Kong University of Science and Technology (Guangzhou)
♣ CSE, The Hong Kong University of Science and Technology
♠ Guangzhou HKUST Fok Ying Tung Research Institute
◇ Terminus Technologies
{xliu886,pdong212,zli755}@connect.hkust-gz.edu.cn
{zhtang.ml, bli}@cse.ust.hk   {xuminghu, xwchu}@hkust-gz.edu.cn

## Abstract

Large Language Models (LLMs) require significant GPU memory when processing long texts, with the key value (KV) cache consuming up to 70% of total memory during inference. Although existing compression methods reduce memory by evaluating the importance of individual tokens, they overlook critical semantic relationships between tokens, resulting in fragmented context and degraded performance. We introduce ChunkKV, which fundamentally reimagines KV cache compression by treating semantic chunks - rather than isolated tokens - as basic compression units. This approach preserves complete linguistic structures and contextual integrity, ensuring that essential meaning is retained even under aggressive compression. Our innovation includes a novel layer-wise index reuse technique that exploits the higher cross-layer similarity of preserved indices in ChunkKV, reducing computational overhead and improving throughput by 26.5%. Comprehensive evaluations on challenging benchmarks: LongBench, Needle-In-A-HayStack, GSM8K, and JailbreakV demonstrate that ChunkKV outperforms state-of-the-art methods by up to 8.7% in precision while maintaining the same compression ratio. These results confirm that semantic-aware compression significantly enhances both efficiency and performance for long-context LLM inference, providing a simple yet effective solution to the memory bottleneck problem. *The code is available at link.*

## 1   Introduction

Large Language Models (LLMs) have become essential for addressing various downstream tasks of natural language processing (NLP), including summarization and question answering, which require the interpretation of a long context from sources such as books, reports, and documents, often encompassing tens of thousands of tokens [1–5]. Recent advances in long-context technology within the field of machine learning (ML) systems [6–8] have significantly improved computational throughputs and reduced latency of LLMs to process increasingly large input context lengths [9, 10] with historical KV cache (key value attentions). However, the memory requirement of the KV cache in serving super-long contexts becomes a new bottleneck [11–14]. For instance, the KV cache for a

---

∗Equal Contribution.
†Corresponding Author.

39th Conference on Neural Information Processing Systems (NeurIPS 2025).

single token in a 7B-parameter model requires approximately 0.5 MB of GPU memory, resulting in a 10,000-token prompt consuming around 5 GB of GPU memory.

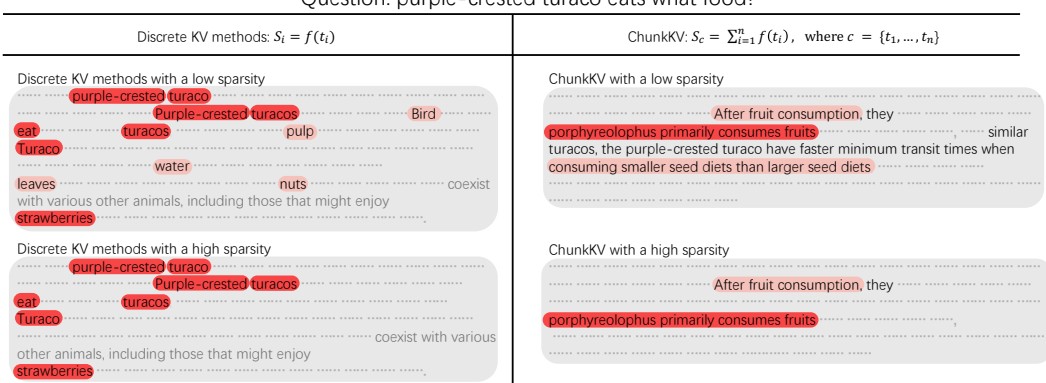

Figure 1: Illustration of the impact of the token discrete method and the chunk method on semantic preservation. The discrete method preserves words related to the question but often omits the subject. In contrast, the chunk method retains the subject of the words, maintaining more accurate semantic information. For the equation: $S$ is the score function, and $c$ is a chunk of tokens.

To address the substantial GPU memory consumption caused by KV caching, recent studies consider compressing the KV cache by pruning non-important discrete parts from the prompt tokens [11, 12, 15–21]. H2O [11] and SnapKV [15] have shown that retaining less than 50% of the discrete KV cache can significantly reduce GPU memory usage with minimal impact on performance. However, we identify that the previous KV cache compression methods [11, 17] measure token importance isolatedly, neglecting the dependency between different tokens on the characteristics of the real-world language. For example, as shown in Figure 1, focusing on the importance of the token level can excessively focus on words about subjects "turaco" in the question while omitting crucial information about objects (foods) in the documents, resulting in the loss of essential semantic information. This motivates us to rethink the following question:

*How to avoid isolated token importance measurement and preserve the semantic information in KV cache?*

Table 1: Comparison of Methods on KV Cache Compression.

| Method | KV Cache Compression | Dynamic Policy | Layer-Wise Policy | Semantic Information | Efficient Index Reuse |
|---|---|---|---|---|---|
| StreamingLLM [8] | ✓ | | | | |
| H2O [11] | ✓ | ✓ | | | |
| SnapKV [15] | ✓ | ✓ | | | |
| PyramidInfer [19] | ✓ | ✓ | ✓ | | |
| PyramidKV [17] | ✓ | ✓ | ✓ | | |
| ChunkKV(Ours) | ✓ | ✓ | ✓ | ✓ | ✓ |

In light of this, we observe that complete semantic information usually appears in a continuous sequence [4, 22–24]. Thus, we introduce a straightforward yet effective ChunkKV, grouping the tokens in a chunk as a basic compressing unit, which should be preserved or discarded as a whole. Thus, it retains the most informative **semantic chunks** from the original KV cache. As shown in Figure 1, preserving a chunk helps catch the subject, predicate, and object. In Table 13, we quantify the minimal loss and higher recovery of the attention score achieved by the chunk-based approach at the inference stage. Furthermore, we investigate that *the preserved KV cache indices by ChunkKV exhibit a higher similarity* compared to previous methods. Consequently, we develop a technique called layer-wise index reuse, which reduces the additional computational time introduced by the KV cache compression method. As outlined in Table 1, recent highly relevant KV cache compression methods *lack the ability to retain semantic information and efficiently reuse indices*.

To evaluate ChunkKV's performance, we conduct comprehensive experiments across multiple cutting-edge long-context benchmarks: long-context tasks including LongBench [25] and Needle-

In-A-HayStack (NIAH) [26], in-context learning tasks such as GSM8K [27] and JailbreakV [28]. And also different models including DeepSeek-R1-Distill-Llama-8B [29],LLaMA-3-8B-Instruct [30], Mistral-7B-Instruct [31], and Qwen2-7B-Instruct [32]. Our experimental results demonstrate that ChunkKV surpasses existing KV cache compression methods in both efficiency and accuracy, primarily due to its ability to preserve essential information through selective chunk retention. These findings establish ChunkKV as a simple yet effective approach to KV cache compression.

We summarize our key contributions as follows:

- We identify the phenomenon in which discrete KV cache compression methods inadvertently prune the necessary semantic information.
- We propose ChunkKV, a simple KV cache compression method that uses the fragmentation method that retains semantic information, and design the layer-wise index reuse technique to reduce the additional computational time.
- We evaluate ChunkKV on cutting-edge long-context benchmarks including LongBench and Needle-In-A-HayStack, as well as the GSM8K, many-shot GSM8K and JailbreakV in-context learning benchmark, and multi-step reasoning (O1 and R1) LLMs, achieving state-of-the-art performance.

## 2 Related Work

**KV Cache Compression.** KV cache compression technology has developed rapidly in the era of LLM, with methods mainly focused on evicting unimportant tokens. The compression process occurs before the attention blocks, optimizing both the prefilling time and GPU memory. Xiao et al. [8] and Han et al. [33] propose that initial and recent tokens consistently have high attention scores between different layers and attention heads. As a result, retaining these tokens in the KV cache is more likely to preserve important information. Furthermore, FastGen [16] evicts tokens based on observed patterns. H2O [11] and SnapKV [15] employ dynamic KV cache compression methods, evaluating the importance of tokens based on attention scores and then evicting the less important ones. As inference scenarios become increasingly complex, dynamic KV cache compression methods demonstrate powerful performance. Recently, Yang et al. [19] and Cai et al. [17] have closely examined the distributions of attention scores during the pre-filling stage of the Retrieval-Augmented Generation (RAG) task, discovering a pyramidal KV cache compression pattern in different transformer layers. In contrast, FlowKV [34] employs a novel multi-turn isolation mechanism that preserves the accumulated compressed KV cache from past turns and only compresses the KV pairs from the most recent turn, effectively mitigating the re-compression of older context and the problem of catastrophic forgetting.

Although these KV cache compression methods have explored efficient GPU memory management while maintaining original performance, our study focuses more on the semantic information of the prompt. We find that chunks of the original KV cache are more important than discrete tokens.

## 3 ChunkKV

### 3.1 Preliminary Study of KV Cache Compression

With the increasing long-context capabilities of LLMs, the KV cache has become crucial for improving the inference efficiency. However, it can consume significant GPU memory when handling long input contexts. The GPU memory cost of the KV cache for the decoding stage can be calculated as follows:

$$M_{KV} = 2 \times B \times S \times L \times N \times D \times 2 \tag{1}$$

where $B$ is the batch size, $S$ is the sequence length of prompt and decoded length, $L$ is the number of layers, $N$ is the number of attention heads, $D$ is the dimension of each attention head, and the first 2 accounts for the KV matrices, while the last 2 accounts for the precision when using float16. Table E shows the configuration parameters for LLaMA-3-8B-Instruct [30]. With a batch size $B = 1$ and a sequence length of prompt $S = 2048$, the GPU memory cost of the KV cache is nearly 1 GB. If the batch size exceeds 24, the GPU memory cost of the KV cache will exceed the capacity of an RTX 4090 GPU. To address this issue, KV cache compression methods have been proposed, with the aim of retaining only a minimal amount of KV cache while preserving as much information as possible. For more details on the LLM configuration parameters, refer to Appendix E.

## 3.2 Chunk Based KV Compression

To address the limitations of existing KV cache compression methods, we propose ChunkKV, a novel KV cache compression method that retains the most informative semantic chunks. The key idea behind ChunkKV is to group tokens in the KV cache into chunks that preserve more semantic information, such as a chunk containing a subject, verb, and object. As illustrated in Figure 1, ChunkKV preserves the chunks of the KV cache that contain more semantic information. First, we define a chunk as a group of tokens that contain related semantic information. By retaining the most informative chunks from the original KV cache, ChunkKV can effectively reduce the memory usage of the KV cache while preserving essential information. For more information on ChunkKV, please refer to Section A.

---

**Algorithm 1** ChunkKV

> **Input:** $\mathbf{Q} \in \mathbb{R}^{T_q \times d}$, $\mathbf{K} \in \mathbb{R}^{T_k \times d}$, $\mathbf{V} \in \mathbb{R}^{T_v \times d}$, observe window size $w$, chunk size $c$, compressed KV cache max length $L_{\max}$
> **Output:** Compressed KV cache $\mathbf{K}'$, $\mathbf{V}'$
> **Observe Window Calculation:**
> $\mathbf{A} \leftarrow \mathbf{Q}_{T_q-w:T_q} \mathbf{K}^T$          ▷ Attention scores for the observe window
> $C \leftarrow \left\lceil \frac{T_k}{c} \right\rceil$          ▷ Calculate the number of chunks
> **Chunk Attention Score Calculation**:
> **for** $i = 1$ to $C$ **do**
>      $\mathbf{A}_i \leftarrow \sum_{j=(i-1)c+1}^{ic} \mathbf{A}_{:,j}$          ▷ Sum of attention scores for each chunk
> **end for**
> **Top-K Chunk Selection**:
> $k \leftarrow \left\lfloor \frac{L_{\max}}{c} \right\rfloor$
> *Top_K_Indices* $\leftarrow$ indices of Top-$k$ chunks based on $\mathbf{A}_i$
> **Compression**:
> $\mathbf{K}', \mathbf{V}' \leftarrow$ index_select($\mathbf{K}, \mathbf{V}, $ *Top_K_Indices*)
> **Concatenation**:
> $\mathbf{K}' \leftarrow$ concat($\mathbf{K}'_{0:L_{\max}-w}, \mathbf{K}_{T_k-w:T_k}$)
> $\mathbf{V}' \leftarrow$ concat($\mathbf{V}'_{0:L_{\max}-w}, \mathbf{V}_{T_v-w:T_v}$)
> $\mathbf{K}', \mathbf{V}'$

---

The algorithm 1 shows the pseudocode for ChunkKV. First, following the approach of H2O [11] and SnapKV [15], we set the observe window by computing the attention scores $\mathbf{A} \leftarrow \mathbf{Q}_{T_q-w:T_q}\mathbf{K}^T$, where $\mathbf{Q}_{T_q-w:T_q}$ is the observe window, $\mathbf{K}$ is the Key matrix and the window size $w$ is usually set to $\{4, 8, 16, 32\}$. Next, the number of chunks $C$ is calculated as $C = \left\lceil \frac{T_k}{c} \right\rceil$, where $T_k$ is the length of the Key matrix and $c$ is the chunk size. The attention scores for each chunk are then computed as $\mathbf{A}_i = \sum_{j=(i-1)c+1}^{ic} \mathbf{A}_{:,j}$ for $i = 1, 2, \ldots, C$. We use the top-$k$ algorithm as the sampling policy for ChunkKV. The top-$k$ chunks are selected based on their attention scores, where $k = \left\lfloor \frac{L_{\max}}{c} \right\rfloor$, and $L_{\max}$ is the maximum length of the compressed KV cache. The size of the last chunk is set to $\min(c, L_{\max} - (k-1) \times c)$. The indices of the top-$k$ chunks preserve the original sequence order. In the compression step, only the key and value matrices corresponding to the selected indices are retained, resulting in the compressed KV cache. Finally, the observe window of the original KV cache will be concatenated to the compressed KV cache by replacing the last $w$ tokens to keep important information. The compressed KV cache is then used for subsequent attention computations. For implementation, we add vectorized operations, memory optimizations, etc., to optimize the code. Refer to Appendix A.5 for more details.

## 3.3 Layer-Wise Index Reuse

Furthermore, we investigated the KV cache indices preserved by ChunkKV and found that they exhibit higher similarity compared to previous methods. Figure 2 shows the layer-wise similarity heatmaps of SnapKV and ChunkKV. Each cell represents the similarity between the

Table 2: Retained KV Cache Indices Similarity of Adjacent Layers for Different Models.

| Method | H2O | SnapKV | ChunkKV |
|---|---|---|---|
| LLaMA-3-8B | 25.31% | 27.95% | **57.74%** |
| Qwen2-7B | 14.91% | 16.50% | **44.26%** |
| Mistral-7B | 15.15% | 15.78% | **52.16%** |

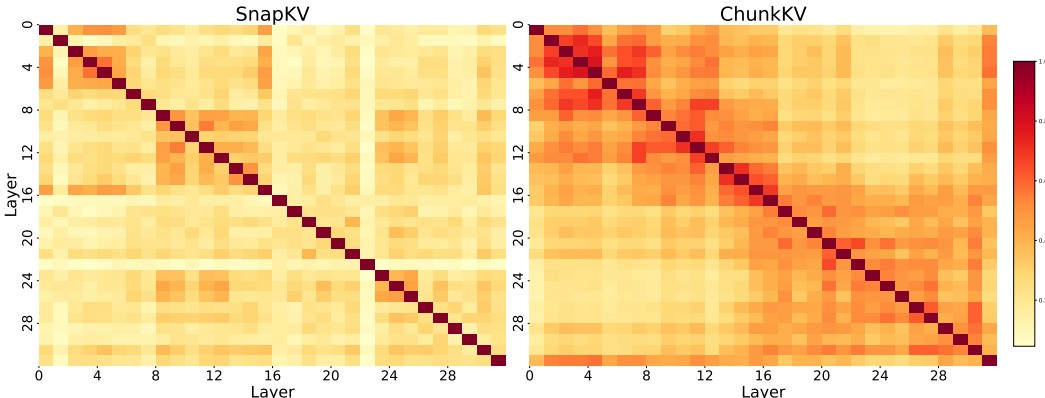

Figure 2: Layer-wise similarity heatmaps of the preserved KV **cache indices** by SnapKV (left) and ChunkKV (right) on LLaMA-3-8B-Instruct. Deep colors indicate higher similarity. More visualization can be found in Appendix B.1.3.

preserved KV cache indices of two layers, with deeper colors indicating higher similarity. The results demonstrate that the KV cache indices preserved by ChunkKV are more similar to those in neighboring layers.

As shown in Table 2, ChunkKV consistently achieves a higher average Jaccard similarity between adjacent layers compared to SnapKV in different model architectures, indicating that the retained token index in ChunkKV is more similar to each other. For a more detailed visualization, please refer to Appendix B.1.3.

Based on the above findings of the KV cache index, we propose a training-free *layer-wise index reuse* method to further reduce the additional cost of KV cache compression, which reuses compressed token indices across multiple layers. We evaluate the efficiency and effectiveness of the layer-wise index reuse method in Section 4.3. Reuse of the layer-wise index reduces the KV cache compression time by 20% compared to the FullKV baseline, with a performance drop of only 0.5%.

This *layer-wise index reuse* method is formally described in Algorithm 2. The ChunkKV compression process returns the compressed KV cache and their respective token indices, denoted as $\mathcal{I}_l$. For layer-wise index reuse, we define a grouping of layers such that all $N_{\text{reuse}}$ layers share the same token indices for ChunkKV. Specifically, for a group of layers $\{l, l+1, \ldots, l+N_{\text{reuse}}-1\}$, we perform ChunkKV on the first layer $l$ to obtain the token indices $\mathcal{I}_l$ and reuse $\mathcal{I}_l$ for the subsequent layers $l+1, l+2, \ldots, l+N_{\text{reuse}}-1$. The notation $\mathbf{K}_l[\mathcal{I}_l]$ and $\mathbf{V}_l[\mathcal{I}_l]$ indicates the selection of key and value caches based on the indices in $\mathcal{I}_l$. The efficiency analysis for layer-wise index reuse is provided in Appendix B.1.1.

**Theoretical Understanding.** We provide a theoretical understanding from the in-context

---

**Algorithm 2** Layer-wise Index Reuse for ChunkKV

**Input:** Number of layers in LLMs $N_{\text{layers}}$, number of reuse layers $N_{\text{reuse}}$
**Initialize:** Dictionary to store indices $\mathcal{I}_{\text{reuse}} = \{\}$
**for** $l = 0$ to $(N_{\text{layers}} - 1)$ **do**
  **if** $l \mod N_{\text{reuse}} == 0$ **then**
    $\mathbf{K}'_l, \mathbf{V}'_l, \mathcal{I}_l \leftarrow \text{ChunkKV}(\mathbf{K}_l, \mathbf{V}_l)$
    $\mathcal{I}_{\text{reuse}}[l] \leftarrow \mathcal{I}_l$
  **else**
    $\mathcal{I}_l \leftarrow \mathcal{I}_{\text{reuse}}\left[\left\lfloor \frac{l}{N_{\text{reuse}}} \right\rfloor \times N_{\text{reuse}}\right]$
  **end if**
  $\mathbf{K}'_l \leftarrow \text{index\_select}(\mathbf{K}_l, \mathcal{I}_l)$
  $\mathbf{V}'_l \leftarrow \text{index\_select}(\mathbf{V}_l, \mathcal{I}_l)$
**end for**

---

Table 3: GSM8K Performance Comparison.
\# SLM=StreamingLLM, SKV=SnapKV,
PKV=PyramidKV

| Ratio | SLM | H2O | SKV | PKV | ChunkKV (Ours) |
|---|---|---|---|---|---|
| DeepSeek-R1-Distill-Llama-8B FullKV: 69.4% ↑ | | | | | |
| 10% | 51.6% | 55.6% | 57.6% | 62.6% | **65.7%** |
| LlaMa-3.1-8B-Instruct FullKV: 79.5% ↑ | | | | | |
| 30% | 70.5% | 72.2% | 76.1% | 77.1% | **77.3%** |
| 20% | 63.8% | 64.0% | 68.8% | 71.4% | **77.6%** |
| 10% | 47.8% | 45.0% | 50.3% | 48.2% | **65.7%** |
| LlaMa-3-8B-Instruct FullKV: 76.8% ↑ | | | | | |
| 30% | 70.6% | 73.6% | 70.2% | 68.2% | **74.6%** |
| Qwen2-7B-Instruct FullKV: 71.1% ↑ | | | | | |
| 30% | 70.8% | 61.2% | 70.8% | 64.7% | **73.5%** |

learning (ICL) [24] to interpret why maintaining the KV cache according to a continuous sequence in ChunkKV is better than according to sparse tokens. Informally speaking, the continuously chunk-level KV cache preserves the whole examples (semantic information) in ICL, thus reducing the requirement on distinguishability, i.e., lower bound of KL divergence between the example and the question (Equation 4 in condition 2). The complete analysis is provided in Appendix C.

## 4  Experiment Results

In this section, we conduct experiments to evaluate the effectiveness of ChunkKV on KV cache compression in two benchmark fields, with a chunk size set to 10 even for various model architectures. The first is the In-Context Learning benchmark, for which we select GSM8K [27] and JailbreakV [28, 35, 36] to evaluate the performance of ChunkKV, furthermore, we also include multi-step reasoning LLM DeepSeek-R1-Distill-Llama-8B [29] to evaluate the performance of ChunkKV. The In-Context Learning scenario is a crucial capability for LLMs and has been adapted in many powerful technologies such as Chain-of-Thought [37–40]. The second is the Long-Context benchmark, which includes LongBench [25] and Needle-In-A-HayStack (NIAH) [26], both widely used for assessing KV cache compression methods. All experiments were carried out three times, using the mean score to ensure robustness.

### 4.1  In-Context Learning

The In-Context Learning (ICL) ability significantly enhances the impact of prompts on LLMs. For example, the Chain-of-Thought approach [37] increases the accuracy of the GSM8K of the PaLM model [41] from 18% to 57% without additional training. In this section, we evaluate the performance of ChunkKV on the GSM8K, Many-Shot GSM8K [42], and JailbreakV [28] benchmarks.

**GSM8K.** In the in-context learning scenario, we evaluated multiple KV cache compression methods for GSM8K [27], which contains more than 1,000 arithmetic questions on LLaMA-3-8B-Instruct, LLaMA-3.1-8B-Instruct [30], Qwen2-7B-Instruct [32] and DeepSeek-R1-Distill-Llama-8B [29]. Following Agarwal et al. [42], we consider many-shot GSM8K as a long-context reasoning scenario, which is a more challenging task than long-context retrieval benchmark Long-

Table 4: Many-Shot (50-shot) GSM8K Performance Comparison.

| Ratio | SLM | H2O | SKV | PKV | ChunkKV (Ours) |
|---|---|---|---|---|---|
| DeepSeek-R1-Distill-Llama-8B FullKV: 71.2% ↑ | | | | | |
| 10% | 63.2% | 54.2% | 54.1% | 59.2% | **68.2%** |
| LlaMa-3.1-8B-Instruct FullKV: 82.4% ↑ | | | | | |
| 10% | 74.3% | 51.2% | 68.2% | 70.3% | **79.3%** |

Bench [25]. The CoT prompt settings for this experiment are the same as those used by Wei et al. [37], for many-shot GSM8K we set the number of shots to 50, where the prompt length is more than 4k tokens. For more details on the prompt settings, please refer to the APPENDIX G.

Table 3 presents the performance comparison. The results show that ChunkKV outperforms other KV cache compression methods on different models and compression ratios. Table 4 presents the performance comparison of many-shot GSM8K, and also ChunkKV outperforms other KV cache compression methods. The consistent superior performance of ChunkKV in both models underscores its effectiveness in maintaining crucial contextual information for complex arithmetic reasoning tasks by chunk level KV cache rather than the discrete token level.

**Jailbreak.** In this section, we evaluate the performance of ChunkKV on the JailbreakV benchmark [28], which is a safety Jailbreak benchmark for language models. The prompt settings are the same as those used by Luo et al. [28].

Table 5: JailbreakV Performance Comparison.

| Ratio | SLM | H2O | SKV | PKV | ChunkKV (Ours) |
|---|---|---|---|---|---|
| LlaMa-3.1-8B-Instruct FullKV: 88.9% ↑ | | | | | |
| 20% | 65.0% | 71.7% | 88.0% | 87.5% | **89.0%** |
| 10% | 53.1% | 65.4% | 84.3% | 85.5% | **87.9%** |

Table 5 presents the performance comparison. The results demonstrate that ChunkKV out-performs other KV cache compression methods on different compression ratios. This shows that for safety benchmark, the chunk level KV cache is more effective than other discrete token level compression methods.

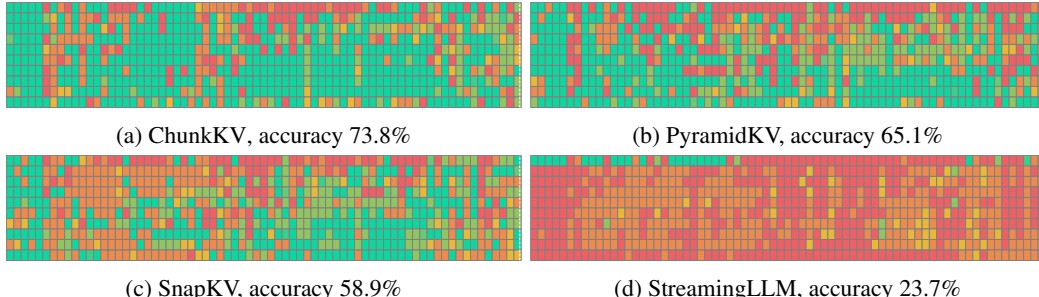

(a) ChunkKV, accuracy 73.8%          (b) PyramidKV, accuracy 65.1%

(c) SnapKV, accuracy 58.9%          (d) StreamingLLM, accuracy 23.7%

Figure 3: NIAH benchmark for LLaMA3-8B-Instruct with KV cache size=128 under 8k context length.

## 4.2 Long-Context Benchmark

LongBench and NIAH are two widely used benchmarks for KV cache compression methods. Both benchmarks have a context length that exceeds $10K$. NIAH requires retrieval capability, while LongBench is a meticulously designed benchmark suite that tests the capabilities of language models in handling extended documents and complex information sequences. For more details on LongBench, please refer to the APPENDIX F.

**LongBench.** We use LongBench [25] to assess the performance of ChunkKV on tasks involving long-context inputs. We evaluated multiple KV cache eviction methods using the LongBench benchmark with LLaMA-3-8B-Instruct [30], Mistral-7B-Instruct-v0.3 [31], and Qwen2-7B-Instruct [32], with a KV cache compression ratio of 10%. LongBench-ZH provides the Chinese subtask, and Qwen2-7B-Instruct also supports Chinese, so we tested Qwen2-7B-Instruct with different KV cache compression methods on the Chinese subtasks.

Tables 6 present the performance gap (in percentage) between each method and the FullKV baseline, where negative values indicate performance degradation compared to FullKV. The table is evaluated in both the LongBench English and Chinese subtasks, where ChunkKV outperforms other compression methods overall. This suggests that ChunkKV's approach of retaining semantic chunks is more effective in preserving important information compared to other discrete token-based compression methods. For the 70B model and Chinese subtask results, please refer to Appendices B.2 and B.5.

**Needle-In-A-HayStack.** We use NIAH [26] to evaluate the long-context retrieval capability of LLMs. NIAH assesses how well LLM extracts hidden tricked information from extensive documents, and following LLM-as-a-Judge [43] we apply GPT-4o-mini [44] to assess the accuracy of the retrieved information. We evaluated multiple KV cache eviction methods using NIAH with LLaMA-3-8B-Instruct and Mistral-7B-Instruct-v0.2, setting benchmark context lengths to 8k and 32k tokens.

Table 6: KV cache compression methods on the LongBench benchmark. Results show performance gap compared to FullKV baseline (negative values indicate worse performance).

| Ratio | SLM | H2O | SKV | PKV | ChunkKV (Ours) |
|---|---|---|---|---|---|
| LlaMa-3-8B-Instruct FullKV: 41.46 ↑ | | | | | |
| 10% | -13.80% | -10.61% | -3.16% | -3.33% | **-2.29%** |
| 20% | -6.42% | -8.85% | -2.24% | -2.00% | **-1.74%** |
| 30% | -2.36% | -5.38% | -0.07% | -0.22% | **+0.31%** |
| Mistral-7B-Instruct-v0.3 FullKV: 48.08 ↑ | | | | | |
| 10% | -16.58% | -9.30% | -3.54% | -3.52% | **-2.85%** |
| Qwen2-7B-Instruct FullKV: 40.71 ↑ | | | | | |
| 10% | -5.28% | -0.64% | -0.39% | -0.98% | **+0.42%** |
| Qwen2-7B-Instruct on LongBench-ZH FullKV: 38.60 ↑ | | | | | |
| 10% | -15.95% | -5.31% | +0.18% | -5.31% | **+2.20%** |

Table 7 provides statistical results for different compression methods. These findings clearly indicate the effectiveness of ChunkKV in managing varying token lengths and depth percentages, making it a robust choice for KV cache management in LLMs. Figure 3 presents the NIAH benchmark results for LLaMA-3-8B-Instruct. The vertical axis represents the depth percentage, while the horizontal axis represents the token length, with shorter lengths on

Table 7: NIAH Performance Comparison.

| KV cache Size | SLM | H2O | SKV | PKV | ChunkKV (Ours) |
|---|---|---|---|---|---|
| LlaMa-3.1-8B-Instruct FullKV: 74.6% ↑ | | | | | |
| 512 | 32.0% | 68.6% | 71.2 % | 72.6% | **74.5%** |
| 256 | 28.0% | 61.7% | 68.8% | 69.5% | **74.1%** |
| 128 | 23.7% | 47.9% | 58.9% | 65.1% | **73.8%** |
| 96 | 21.5% | 41.0% | 56.2% | 63.2% | **70.3%** |
| Mistral-7B-Instruct FullKV: 99.8% ↑ | | | | | |
| 128 | 44.3% | 88.2% | 91.6% | 99.3% | **99.8%** |

the left and longer lengths on the right. A cell highlighted in green indicates that the method can retrieve the needle at that length and depth percentage. The detailed visualization of the NIAH benchmark can be found in the Appendix B.3. The visualization results demonstrate that ChunkKV outperforms other KV cache compression methods.

## 4.3 Index Reuse

This section will evaluate the performance of the layer-wise index reuse approach with ChunkKV from the two aspects of efficiency and performance.

**Measuring Efficiency.** We evaluated the latency and throughput of ChunkKV compared to FullKV using LLaMA3-8B-Instruct on an A40 GPU. All experiments were conducted with reuse layer is 2, batch size set to 1 and inference was performed using Flash Attention 2, each experiment was repeated 10 times and the average latency and throughput were reported.

The results in Table 8 show that the layer-wise index reuse strategy (ChunkKV_reuse) further boosts performance, achieving up to a 20.7% reduction in latency, and throughput improvements are particularly notable for longer input sequences, with ChunkKV_reuse delivering up to a 26.5% improvement over FullKV. For more detailed results, please refer to Appendix B.8.

Table 8: Latency and throughput comparison between ChunkKV and FullKV under different input-output configurations. Percentages in parentheses indicate improvements over FullKV baseline.

| Method | Sequence Length | | Performance Metrics | |
|---|---|---|---|---|
| | Input | Output | Latency(s) ↓ | Throughput(T/S) ↑ |
| FullKV | 4096 | 1024 | 43.60 | 105.92 |
| ChunkKV | 4096 | 1024 | 37.52 (13.9%) | 118.85 (12.2%) |
| ChunkKV_reuse | 4096 | 1024 | **37.35** (14.3%) | **124.09** (17.2%) |
| FullKV | 4096 | 4096 | 175.50 | 37.73 |
| ChunkKV | 4096 | 4096 | 164.55 (6.2%) | 40.58 (7.6%) |
| ChunkKV_reuse | 4096 | 4096 | **162.85** (7.2%) | **41.12** (9.0%) |
| FullKV | 8192 | 1024 | 46.48 | 184.08 |
| ChunkKV | 8192 | 1024 | 37.83 (18.6%) | 228.96 (24.4%) |
| ChunkKV_reuse | 8192 | 1024 | **36.85** (20.7%) | **232.99** (26.5%) |
| FullKV | 8192 | 4096 | 183.42 | 55.93 |
| ChunkKV | 8192 | 4096 | 164.78 (10.2%) | 65.14 (16.5%) |
| ChunkKV_reuse | 8192 | 4096 | **162.15** (11.6%) | **66.05** (18.1%) |

**Measuring Task Performance.** We evaluate the effectiveness of our proposed layer-wise index reuse technique on Long-Bench [25] and GSM8K [27] benchmarks. For these experiments, we use the same configuration as our main LongBench experiments in Section 4.2, with index reuse applied to consecutive layers (reuse layers = 2).

As shown in Table 9, layer-wise index reuse maintains the performance of the models while reducing computational requirements. In LongBench, performance degradation is minimal (less than 0.6%) for both models, while GSM8K shows neutral or slightly positive effects. This validates that semantic chunks selected by ChunkKV remain consistently important across adjacent transformer layers, enabling efficient computation without sacrificing accuracy. An additional analysis of different reuse depths and their impact on throughput is provided in Appendix B.1.2.

Overall, these findings on efficiency and performance suggest that layer-wise index reuse can be an effective technique for optimizing the efficiency-performance trade-off in KV cache compression, with the potential for model-specific tuning to maximize benefits.

Table 9: Reusing Indexing Performance Comparison on LongBench and GSM8K. △ indicates the performance degradation of index reuse compared to the baseline.

| Model | ChunkKV | |
|---|---|---|
| | Baseline | Index Reuse$_\triangle$ |
| LongBench | | |
| LLaMA-3-8B-Inst | 40.51 | 40.27$_{-0.59\%}$ |
| Mistral-7B-Inst | 46.71 | 46.43$_{-0.59\%}$ |
| Qwen2-7B-Inst | 40.88 | 40.76$_{-0.29\%}$ |
| GSM8K | | |
| LLaMA-3-8B-Inst | 74.5 | 74.6$_{+0.13\%}$ |
| Qwen2-7B-Inst | 71.2 | 71.2$_{+0.00\%}$ |

## 4.4 Chunk Size

This section aims to investigate the impact of chunk size on the performance of ChunkKV. Different chunk sizes will lead to varying degrees of compression on the semantic information of the data. We used the same experiment setting as in LongBench and NIAH (Section 4.2). The chunk size is set from the range $\{3, 5, 10, 20, 30\}$ under the compression rate 10% for LongBench and the 128 KV cache size for NIAH. For more experiments with different compression ratios, see Appendix B.4.

Table 10: LongBench and NIAH Results with Chunk Size Ablation

| Model | Full KV | Chunk KV | | | | | SnapKV | H2O |
|---|---|---|---|---|---|---|---|---|
| | | size=3 | size=5 | size=10 | size=20 | size=30 | | |
| **LongBench ↑** | | | | | | | | |
| LLaMA-3-8B-Instruct | 41.46 | 40.49 | 40.47 | **40.51** | 40.05 | 39.57 | 40.15 | 37.06 |
| Mistral-7B-Instruct | 48.08 | 46.45 | 46.51 | **46.71** | 46.42 | 45.98 | 46.38 | 43.61 |
| **NIAH ↑** | | | | | | | | |
| LLaMA-3-8B-Instruct | 74.6 | 65.6 | 69.1 | **73.8** | 72.0 | 71.2 | 58.9 | 47.9 |
| Mistral-7B-Instruct | 99.8 | 98.1 | 99.2 | **99.8** | 99.8 | 99.1 | 91.6 | 88.2 |

As shown in Table 10, the performance remains relatively stable when the chunk size is between 5 and 20, with the best results consistently achieved at chunk size 10. When the chunk size is too small (e.g., 3), the context is fragmented, leading to a slight drop in performance. Conversely, when the chunk size is too large (for example, 30), the semantic granularity becomes too coarse, and important fine-grained information may be lost, also resulting in performance degradation. This trend is consistent across both LongBench and NIAH benchmarks, as well as across different model architectures, indicating that the optimal chunk size is not highly sensitive to the specific task or model. We also provide a line graph in Appendix B.4 to visually illustrate this trend. Based on these findings, we recommend using a chunk size of 10 as a robust default for most applications. For users with specific requirements, chunk size can be further tuned, but our results suggest that moderate values (5-20) generally offer a good trade-off between semantic preservation and compression efficiency.

## 4.5 Comparing with KV Quantization

For comprehensively evaluate the effectiveness of ChunkKV, we performed experiments comparing ChunkKV with the KIVI quantization methods [45]. Although both approaches aim to optimize LLM inference, they operate on fundamentally different principles: Quantization reduces KV matrix precision, whereas our eviction method reduces KV matrix size. For more detail, see Appendix B.6.

From an implementation perspective, quantization methods require the full KV cache during prefilling to produce quantized representations, which are then used during decoding. In contrast, ChunkKV employs token removal prior to prefilling, enabling operation with a compressed cache throughout the entire inference process. This distinction creates different efficiency profiles, and each method offers unique advantages.

Due to the fact that KIVI requires an old Python version, the ChunkKV results are not aligned with Table 8. The efficiency results in Table 11 reveal ChunkKV's significant advantages in latency-critical metrics. Although both methods substantially reduce cache size (ChunkKV to 10% and KIVI-2bits to 15.63%), ChunkKV delivers superior metrics for Time to First Token (TTFT) and Token Processing Time (TPOT). Most remarkably, ChunkKV achieves a 164.66s total generation time compared to KIVI's 226.52s at 2-bit quantization, representing a 27.3% improvement in overall inference speed.

Table 11: Efficiency Results for ChunkKV and KIVI on LLaMa-3-8B-Instruct

| Configuration | Prompt Length | Output Length | Compression Ratio / nbits | Prefilling Time(s) ↓ | Cache Size(GB) ↓ | TTFT (s) ↓ | TPOT (ms) ↓ | Total Gen. Time(s) ↓ |
|---|---|---|---|---|---|---|---|---|
| FullKV | 8192 | 4096 | - | 1.5621 | 1.0000 | 1.6013 | 45.9421 | 184.2934 |
| KIVI | 8192 | 4096 | 2bits | 1.4024 | 0.1563 | 1.4325 | 54.9561 | 226.5234 |
| KIVI | 8192 | 4096 | 4bits | 1.3916 | 0.2813 | 1.4146 | 52.0510 | 214.5634 |
| ChunkKV | 8192 | 4096 | 10% | 1.3653 | 0.1000 | 1.3914 | 39.8702 | 164.6600 |

## 4.6 Analysis of Hybrid Compression: Chunk-level vs. Token-level at Different Layer Depths

A critical question raised during the review process was whether the benefits of chunk-level compression diminish in deeper Transformer layers, where semantic information becomes more abstract and diffused. To investigate this, we designed a hybrid compression model that applies different strategies at varying network depths.

**Experimental Setup** We created a hybrid version of the LLaMA-3-8B-Instruct model. For the first 16 layers (bottom half), we applied our chunk-based ChunkKV. For the final 16 layers (top half), we applied SnapKV, a state-of-the-art token-level compression method. We then compared this hybrid model's performance on the diverse LongBench benchmark against pure ChunkKV and pure SnapKV at identical compression ratios.

Table 12: Performance of the hybrid compression model on LongBench. While the hybrid model shows strengths in global understanding tasks, pure ChunkKV achieves the best overall performance, validating the robustness of preserving semantic chunks even in deep layers.

| Method | Single-Doc QA | Multi-Doc QA | Summarization | Few-shot | Synthetic | Code | Avg. Score ↑ |
|---|---|---|---|---|---|---|---|
| FullKV | 32.19 | 34.59 | 24.96 | 68.48 | 36.96 | 54.41 | 41.46 |
| SnapKV (10% Ratio) | 28.11 | 32.55 | 24.12 | 67.81 | 36.01 | 55.67 | 40.15 |
| Hybrid Model (10% Ratio) | 28.38 | 30.37 | **24.54** | **67.87** | 36.37 | 55.32 | 39.80 |
| ChunkKV (10% Ratio) | **28.50** | **33.46** | 22.20 | 67.62 | **37.47** | **58.98** | **40.51** |

**Analysis and Insights** The results in Table 12 provide two key insights. First, the pure ChunkKV model achieves the highest overall average score. This empirically validates our core hypothesis: preserving local semantic integrity via chunking is a robust and effective strategy across all layers of the model, even where information is highly processed.

Second, the hybrid model reveals a fascinating, task-dependent performance trade-off.

- **For local information retrieval tasks** (e.g., Single- and Multi-Document QA), pure ChunkKV is significantly superior. These tasks often require retrieving precise, intact text fragments. ChunkKV's ability to preserve complete linguistic units prevents critical information from being fragmented, which is essential in these scenarios.
- **For global understanding tasks** (e.g., Summarization and Few-shot Learning), the hybrid model performs best. In these tasks, synthesizing information from across the entire context is key. In the deeper layers, where abstract representations are formed, the token-level SnapKV method may be more adept at retaining a broader, more diffuse set of globally important signals.

This analysis does not undermine our approach but rather enriches it, suggesting that the future of KV cache compression may lie in adaptive, task-aware strategies. However, for a general-purpose and robust solution, pure ChunkKV proves to be the most effective.

## 5 Conclusion

In this paper, we first indicate that current KV cache methods lack the semantic information of the data, and then we proposed a novel KV cache compression method that preserves semantic information by retaining more informative chunks. Through extensive experiments across multiple state-of-the-art LLMs (including DeepSeek-R1, LLaMA-3, Qwen2, and Mistral) and diverse benchmarks (GSM8K, LongBench, NIAH, and JailbreakV), we demonstrate that ChunkKV consistently outperforms existing methods while using only a fraction of the memory. Our comprehensive analysis shows that ChunkKV's chunk-based approach maintains crucial contextual information, leading to superior performance in complex reasoning tasks, long-context understanding, and safety evaluations. The method's effectiveness is particularly evident in challenging scenarios like many-shot GSM8K and multi-document QA tasks, where semantic coherence is crucial. Furthermore, our proposed layer-wise index reuse technique provides significant computational efficiency gains with minimal performance impact, achieving up to 20.7% latency reduction and 26.5% throughput improvement. These findings, supported by detailed quantitative analysis and ablation studies, establish ChunkKV as a significant advancement in KV cache compression technology, offering an effective solution for deploying LLMs in resource-constrained environments while maintaining high-quality outputs.

## Acknowledge

This work was supported by the NSFC grant 62432008; RGC RIF grant R6021-20; an RGC TRS grant T43-513/23N-2; RGC CRF grants C7004-22G, C1029-22G and C6015-23G; NSFC/RGC grant CRS_HKUST601/24 and RGC GRF grants 16207922, 16207423 and 16203824; National

Natural Science Foundation of China (Grant No.62506318); Guangdong Provincial Department of Education Project (Grant No.2024KQNCX028); Scientific Research Projects for the Higher-educational Institutions (Grant No.2024312096), Education Bureau of Guangzhou Municipality; Guangzhou-HKUST(GZ) Joint Funding Program (Grant No.2025A03J3957), Education Bureau of Guangzhou Municipality;the Guangzhou Municipal Joint Funding Project with Universities and Enterprises under Grant No. 2024A03J0616 and Guangzhou Municipality Big Data Intelligence Key Lab (2023A03J0012).

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

# Appendix

# A   In-depth Analysis of ChunkKV vs. Discrete Token Methods

## A.1   Quantitative Analysis

To rigorously evaluate the effectiveness of ChunkKV compared to discrete token-based methods, we conducted systematic experiments using a LLaMA-3-8B-Instruct model. We randomly selected 100 sequences from the each sub-category of LongBench dataset and analyzed two key metrics across different model layers: KV cache L1 loss and attention cosine similarity. For each sequence, we: 1. Computed the full KV cache and attention patterns without compression as ground truth. 2. Applied ChunkKV, SnapKV, and H2O compression methods with a fixed 10% compression ratio, and the parameters of the three methods are set the same as in Table 17. 3. Measured the differences between compressed and uncompressed versions.

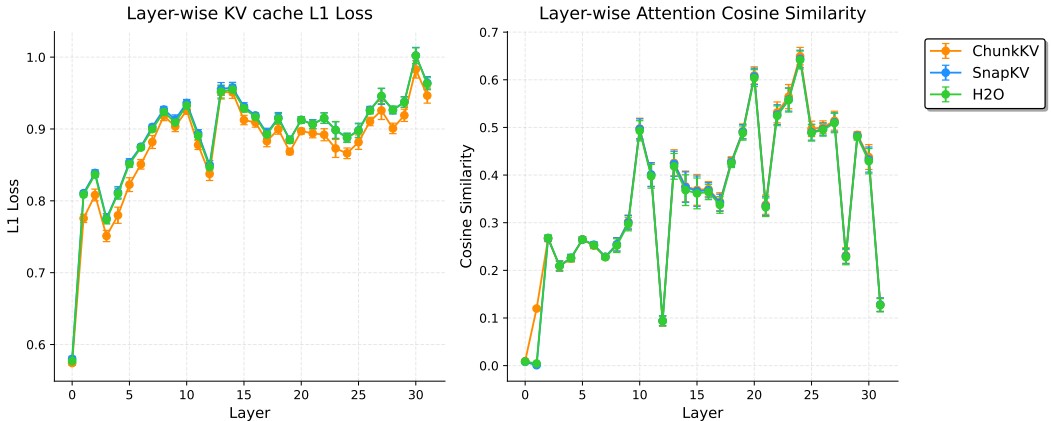

Figure 4: Layer-wise comparison of L1 loss and attention cosine similarity between ChunkKV and discrete token-based methods in Single-Document QA sub-category of LongBench.

**Results Analysis** As shown in Figure 4, ChunkKV demonstrates superior performance across both metrics:

- **KV Cache L1 Loss:** ChunkKV achieves consistently lower L1 loss compared to SnapKV and H2O, particularly in the early and middle layers (layers 5-25). This indicates better preservation of the original KV cache information through the semantic chunk-based approach.

- **Attention Cosine Similarity:** ChunkKV exhibits higher similarity scores across most layers, with notably strong performance in layers 0-5 and 20-30. This suggests better preservation of attention relationships between tokens, which is crucial for maintaining semantic understanding.

To quantify these improvements, we calculated average metrics across all layers, as shown in Table 13. ChunkKV achieves both the lowest L1 loss and highest attention cosine similarity, outperforming both baseline methods.

**Significance of Results** While the improvements may appear modest in absolute terms (approximately 2% in L1 loss and 1.5% in cosine similarity), their practical significance is substantial. These metrics reflect the model's ability to maintain crucial semantic relationships and attention patterns, which are essential for complex reasoning tasks. The consistent improvements across different sequences demonstrate that preserving semantic chunks leads to better information retention than selecting individual tokens.

The enhanced performance is particularly evident in the middle layers of the model, which are typically responsible for higher-level semantic processing. This provides concrete evidence for why ChunkKV achieves superior performance on downstream tasks compared to discrete token-based methods.

Table 13: Detailed comparison of KV cache metrics across different task categories in LongBench.

| Method | Single-Document QA | Multi-Document QA | Summarization | Few-shot Learning | Synthetic & Code |
|---|---|---|---|---|---|
| KV Cache L1 Loss ↓ | | | | | |
| **ChunkKV** | **0.8741** | **0.8748** | **0.8770** | **0.8861** | **0.8726** |
| SnapKV | 0.8921 | 0.8933 | 0.8930 | 0.8917 | 0.8938 |
| H2O | 0.8905 | 0.8917 | 0.8913 | 0.8906 | 0.8915 |
| Attention Score Cosine Similarity ↑ | | | | | |
| **ChunkKV** | **0.3567** | **0.3651** | **0.3841** | **0.4330** | **0.3805** |
| SnapKV | 0.3513 | 0.3594 | 0.3771 | 0.4305 | 0.3759 |
| H2O | 0.3491 | 0.3572 | 0.3750 | 0.4284 | 0.3740 |

## A.2 Hypothetical Scenario

To provide a deeper understanding of ChunkKV's effectiveness compared to discrete token-based methods, we present a detailed analysis using a hypothetical scenario. This analysis aims to illustrate the fundamental differences between these approaches and explain why ChunkKV is more effective at preserving semantic information in long contexts.

Consider a comprehensive document that contains detailed information on various animals, including their habitats, diets, and behaviors. A user asks the question "What do pandas eat in the wild?"

Both ChunkKV and discrete token-based methods would use this question to calculate attention scores for the document. However, their approaches to selecting and retaining information differ significantly.

### A.2.1 Discrete Token-based Method

A discrete token-based method might identify and retain individual tokens with high relevance scores, such as:

- "pandas","eat", "bamboo", "wild", "diet", "food"

Although these tokens are relevant, they lack context and coherence. The method might discard other essential tokens that provide crucial context or complete the information.

### A.2.2 ChunkKV Method

In contrast, ChunkKV would identify and retain semantically meaningful chunks, such as:

- "In the wild, pandas primarily eat bamboo shoots and leaves"
- "Their diet consists of 99% bamboo, but they occasionally consume other vegetation"
- "Wild pandas may also eat small rodents or birds when available"

By preserving these chunks, ChunkKV maintains not only the relevant keywords but also their contextual relationships and additional pertinent information.

### A.3 Comparative Analysis

The advantages of ChunkKV become evident when we consider how these retained pieces of information would be used in subsequent processing:

1. **Contextual Understanding**: Discrete tokens require the model to reconstruct meaning from isolated words, which could lead to ambiguity. ChunkKV provides complete phrases or sentences, allowing for immediate and accurate comprehension.

2. **Semantic Coherence**: ChunkKV preserves the semantic relationships within a chunk, crucial to understanding nuances such as the difference between primary and occasional food sources for pandas.

3. **Information Density**: A single chunk can contain multiple relevant tokens in their proper context, potentially retaining more useful information within the same compressed cache size compared to discrete methods.

4. **Reduced Ambiguity**: Discrete methods might retain the token "eat" from various sentences about different animals. ChunkKV ensures that "eat" is preserved specifically in the context of pandas in the wild.

5. **Temporal and Logical Flow**: ChunkKV can maintain the sequence of ideas present in the original text, preserving any temporal or logical progression that may be crucial for understanding.

### A.4 Implications for Model Performance

This analysis suggests several key implications for model performance:

- **Improved Accuracy**: By retaining contextually rich information, ChunkKV enables more accurate responses to queries, especially those requiring nuanced understanding.
- **Enhanced Long-context Processing**: Preservation of semantic chunks allows for better handling of long-range dependencies and complex reasoning tasks.
- **Reduced Computational Overhead**: Although both methods compress the KV cache, ChunkKV's approach may reduce the need for extensive context reconstruction, potentially improving inference efficiency.
- **Versatility**: The chunk-based approach is likely to be more effective across a wide range of tasks and domains as it preserves the natural structure of language.

This in-depth analysis demonstrates why ChunkKV is more effective in preserving semantic information in long contexts. By retaining coherent chunks of text, it provides language models with more contextually rich and semantically complete information, leading to improved performance in tasks that require deep understanding and accurate information retrieval from extensive documents.

### A.5 Implementation Details

In our implementation of ChunkKV, we focus on maximizing computational efficiency to ensure practical deployment in real-world applications. In algorithm 3, we adopt several key optimization strategies to reduce both computational overhead and memory footprint during inference.

The optimized implementation features several key improvements:

**Vectorized Operations.** We leverage batch matrix operations for attention score calculation instead of explicit loops, significantly reducing computational overhead. By computing chunk scores through optimized tensor operations (lines 7-11), we achieve substantial acceleration compared to token-by-token processing.

**Memory Optimization.** Our implementation employs a binary mask approach (lines 15-20) that avoids redundant memory allocations and reduces fragmentation. This single-pass token selection strategy consolidates both chunk preservation and recent window retention operations, leading to more efficient memory usage during inference.

**Boundary Handling.** We incorporate robust boundary checks throughout the algorithm (lines 10, 12, 18) to handle edge cases such as incomplete chunks or limited compressed length. This ensures stable performance across variable input lengths without requiring special case handling.

**Single-pass Compression.** Rather than performing multiple concatenation operations, our implementation builds a comprehensive selection mask that includes both semantically important chunks and recent context tokens. This approach (lines 21-25) minimizes memory copying operations and reduces computational overhead.

Our PyTorch implementation further extends these optimizations by leveraging GPU acceleration for all vector and matrix operations. The integration of CUDA kernels for critical operations like

---

**Algorithm 3** ChunkKV Implementation

---

1: **Input:** $\mathbf{Q} \in \mathbb{R}^{T_q \times d}$, $\mathbf{K} \in \mathbb{R}^{T_k \times d}$, $\mathbf{V} \in \mathbb{R}^{T_v \times d}$, observe window size $w$, chunk size $c$, compressed KV cache max length $L_{\max}$
2: **Output:** Compressed KV cache $\mathbf{K}'$, $\mathbf{V}'$
3: $q_{\text{observe}} \leftarrow \mathbf{Q}_{T_q - w : T_q}$        ▷ Extract observe window queries
4: $\mathbf{A} \leftarrow q_{\text{observe}} \mathbf{K}^T$        ▷ $\mathbb{R}^{w \times T_k}$ attention scores
5: $C \leftarrow \lceil \frac{T_k}{c} \rceil$        ▷ Number of chunks
6: Allocate chunk_scores $\in \mathbb{R}^C$ to store chunk importance
7: **Vectorized Chunk Score Calculation:**
8: **for** $i = 0$ to $C - 1$ **do**
9:      start $\leftarrow i \cdot c$
10:     end $\leftarrow \min((i + 1) \cdot c, T_k)$
11:     chunk_scores$[i] \leftarrow \sum_{j=\text{start}}^{\text{end}-1} \sum_{q=0}^{w-1} \mathbf{A}_{q,j}$        ▷ Vectorized sum
12: **end for**
13: $k \leftarrow \min(\lfloor \frac{L_{\max} - w}{c} \rfloor, C)$        ▷ Limit k to available chunks
14: kept_chunks $\leftarrow$ Top-K(chunk_scores, $k$)        ▷ Indices of top-k chunks
15: **Build selection mask:**
16: mask $\leftarrow$ zeros($T_k$)        ▷ Binary mask for token selection
17: **for** each $i$ in kept_chunks **do**
18:      start $\leftarrow i \cdot c$
19:      end $\leftarrow \min((i + 1) \cdot c, T_k)$
20:      mask[start : end] $\leftarrow 1$        ▷ Mark entire chunk as kept
21: **end for**
22: **Ensure recent context is preserved:**
23: mask$[T_k - w : T_k] \leftarrow 1$        ▷ Always keep most recent tokens
24: **Efficient compression:**
25: indices $\leftarrow$ nonzero(mask)        ▷ Get indices of tokens to keep
26: $\mathbf{K}' \leftarrow$ index_select($\mathbf{K}$, indices)
27: $\mathbf{V}' \leftarrow$ index_select($\mathbf{V}$, indices)
28: **Return** $\mathbf{K}'$, $\mathbf{V}'$

---

attention score calculation and top-k selection enables real-time performance even with long context windows.

## B    Additional Experiments

### B.1    Layer-Wise Index Reuse

#### B.1.1    Efficiency Analysis

The layer-wise index reuse method significantly reduces the computational complexity of ChunkKV. Without index reuse, ChunkKV would be applied to all $N_{\text{layers}}$ layers, resulting in a total compression time of $N_{\text{layers}} \cdot T_{\text{compress}}$, where $T_{\text{compress}}$ is the time taken to compress one layer. With index reuse, ChunkKV is only applied to $\frac{N_{\text{layers}}}{N_{\text{reuse}}}$ layers, reducing the total time to $\frac{N_{\text{layers}}}{N_{\text{reuse}}} \cdot T_{\text{compress}} + (N_{\text{layers}} - \frac{N_{\text{layers}}}{N_{\text{reuse}}}) \cdot T_{\text{select}}$, where $T_{\text{select}}$ is the time taken to select indices, which is typically much smaller than $T_{\text{compress}}$. This results in a theoretical speedup factor of:

$$\text{Speedup} = \frac{N_{\text{layers}} \cdot T_{\text{compress}}}{\frac{N_{\text{layers}}}{N_{\text{reuse}}} \cdot T_{\text{compress}} + (N_{\text{layers}} - \frac{N_{\text{layers}}}{N_{\text{reuse}}}) \cdot T_{\text{select}}}$$

Assuming $T_{\text{select}}$ is negligible compared to $T_{\text{compress}}$, this simplifies to approximately $N_{\text{reuse}}$. In practice, the actual speedup may vary depending on the specific implementation and hardware, but it can still lead to substantial time savings, especially for models with a large number of layers.

## B.1.2  Index Reuse Performance

Figure 5 and Table 14 illustrates the performance of ChunkKV with varying index reuse layers on the GSM8K benchmark. Figure 6 and Table 15 illustrates the performance of ChunkKV with varying index reuse layers on the LongBench benchmark. The experiment reveals that math problems are more sensitive to index reuse layers compared to LongBench. Both LLaMA3-8B-Instruct and Qwen2-7B-Instruct exhibit significant performance degradation, with LLaMA3-8B-Instruct experiencing a steeper decline after two layers of index reuse than Qwen2-7B-Instruct. This suggests that the Qwen2-7B-Instruct model may be more robust to index reuse.

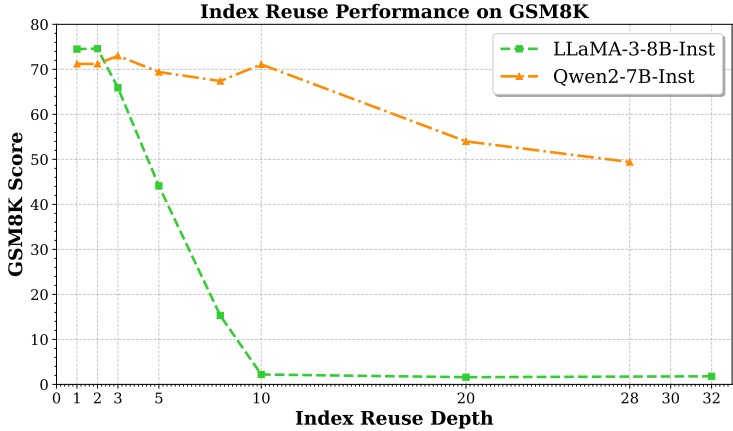

Figure 5: GSM8K Performance Comparison with different index reuse layers

Table 14: Reusing Indexing Performance Comparison on GSM8K

| Model | Number of Index Reuse Layers | | | | | | | |
|---|---|---|---|---|---|---|---|---|
| | 1 | 2 | 3 | 5 | 8 | 10 | 20 | 28/32 |
| LLaMA-3-8B-Instruct | 74.5 | 74.6 | 65.9 | 44.1 | 15.3 | 2.20 | 1.60 | 1.80 |
| Qwen2-7B-Instruct | 71.2 | 71.2 | 73.0 | 69.4 | 67.4 | 71.1 | 54.0 | 49.4 |

Table 15: Reusing Indexing Performance Comparison on LongBench

| Model | Number of Index Reuse Layers | | | | | | |
|---|---|---|---|---|---|---|---|
| | 1 | 2 | 3 | 5 | 10 | 20 | 28/32 |
| LLaMA-3-8B-Instruct | 40.51 | 40.27 | 39.45 | 40.05 | 39.59 | 37.42 | 35.67 |
| Mistral-7B-Instruct | 46.71 | 46.43 | 46.48 | 45.90 | 44.36 | 41.84 | 39.56 |
| Qwen2-7B-Instruct | 40.88 | 40.76 | 40.91 | 41.58 | 40.19 | 37.42 | 36.79 |

## B.1.3  Layer-Wise Index Similarity

This section details the experiment of layer-wise index reuse similarity described in Section 3.3. The inference prompt is randomly selected from the LongBench benchmark, and the preserved indices for H2O, SnapKV, and ChunkKV are saved in the log file. For multi-head attention, only the indices of the first head are saved. PyramidKV, which has varying preserved index sizes across different layers, is not applicable for this experiment. Then we calculate the Jaccard similarity of the preserved indices of adjacent layers for different models. Table 16 shows the Jaccard similarity of the preserved indices of adjacent layers for different models.

Figures 7-9 (LLaMA-3-8B-Instruct), 10-12 (Mistral-7B-Instruct), and 13-15 (Qwen2-7B-Instruct) display the heatmaps of layer-wise indices similarity of the preserved KV cache indices by H2O, SnapKV and ChunkKV on different models. The pattern of the layer-wise indices similarity heatmap is consistent across different models, aligning with our findings in Section 3.3.

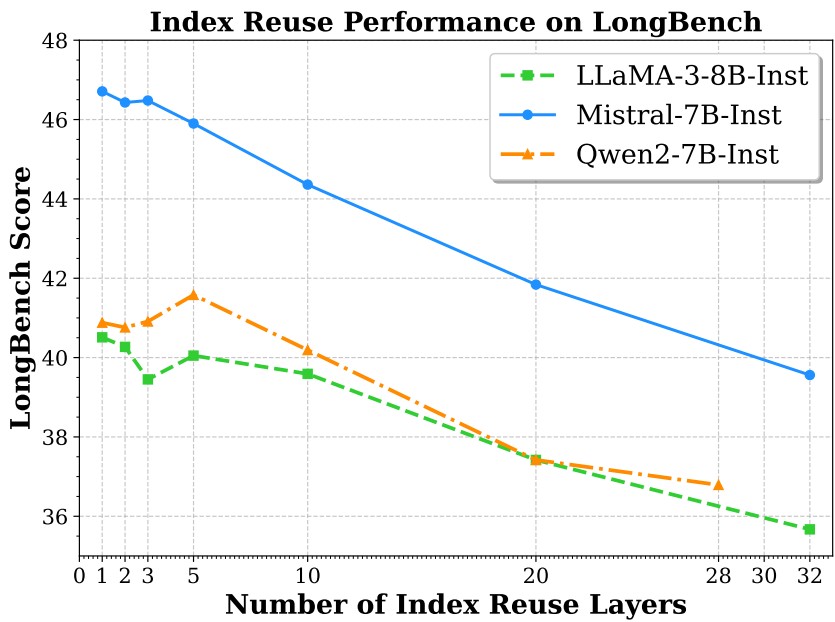

Figure 6: Comparison with different index reuse layers on LongBench.

Table 16: Retained KV Cache Indices Similarity of Adjacent Layers for Different Models.

| Method | H2O | SnapKV | ChunkKV |
|---|---|---|---|
| LLaMA-3-8B-Instruct | 25.31% | 27.95% | **57.74%** |
| Qwen2-7B-Instruct | 14.91% | 16.50% | **44.26%** |
| Mistral-7B-Instruct | 15.15% | 15.78% | **52.16%** |

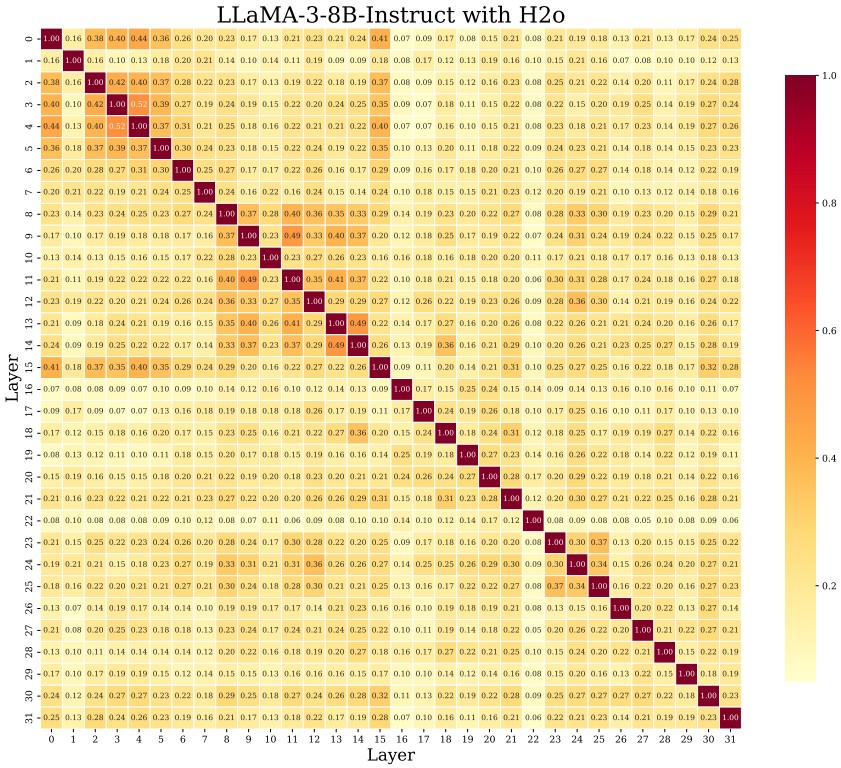

Figure 7: Layer-wise similarity heatmaps of the preserved KV cache indices by H2O on LLaMA-3-8B-Instruct

## B.2 LongBench

The Table 17 shows the average performance of KV cache compression methods in the LongBench English subtask categories. The ChunkKV achieves the best performance on the overall average, and the Multi-Document QA category, which supports that chunk method is more effective for semantic preservation.

Table 18 shows additional experiments with LLaMA-3-70B-Instruct, comparing our ChunkKV method with baselines with KV cache size = 256 on the LongBench dataset:

## B.3 Needle-In-A-Haystack

Figure 16 and 17 visualizes the performance of ChunkKV on the NIAH benchmark for LLaMA-3-8B-Instruct and Mistral-7B-Instruct with a KV cache size of 128 under 8k and 32k context length. The performance of ChunkKV is consistently better as the context length increases.

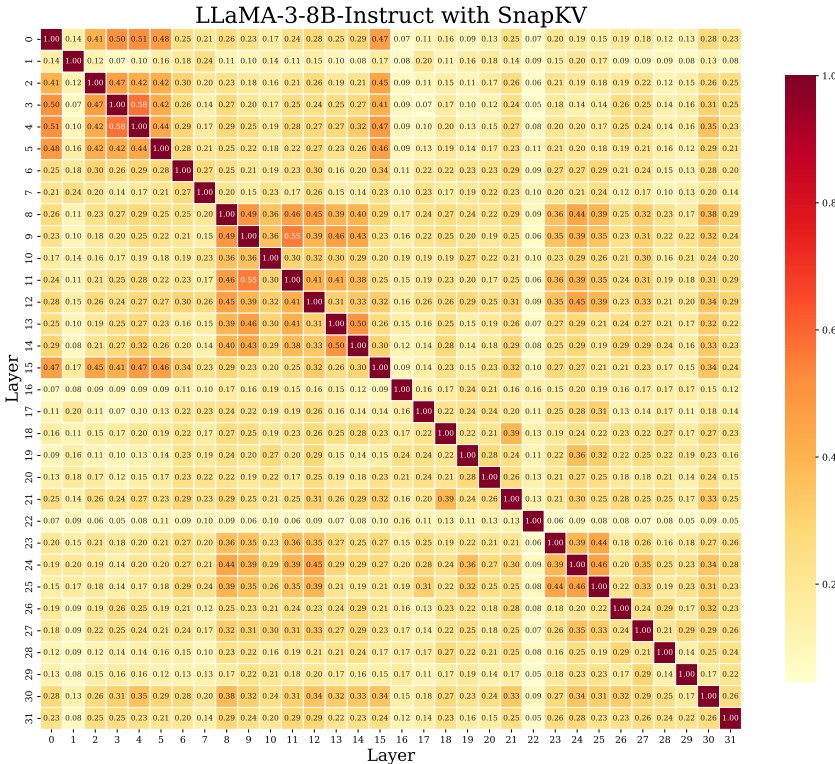

Figure 8: Layer-wise similarity heatmaps of the preserved KV cache indices by SnapKV on LLaMA-3-8B-Instruct

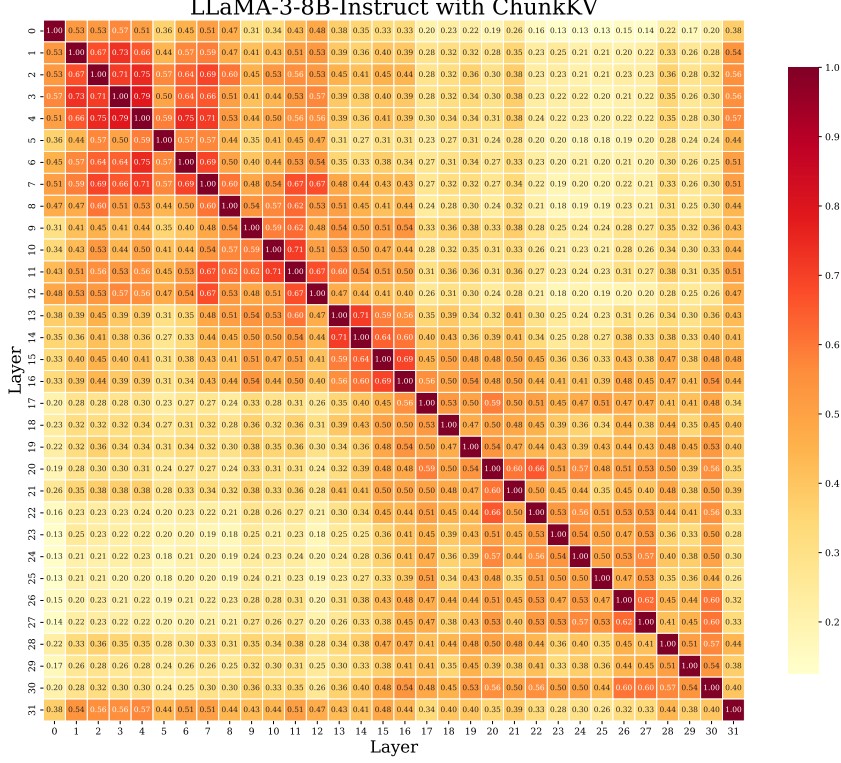

Figure 9: Layer-wise similarity heatmaps of the preserved KV cache indices by ChunkKV on LLaMA-3-8B-Instruct

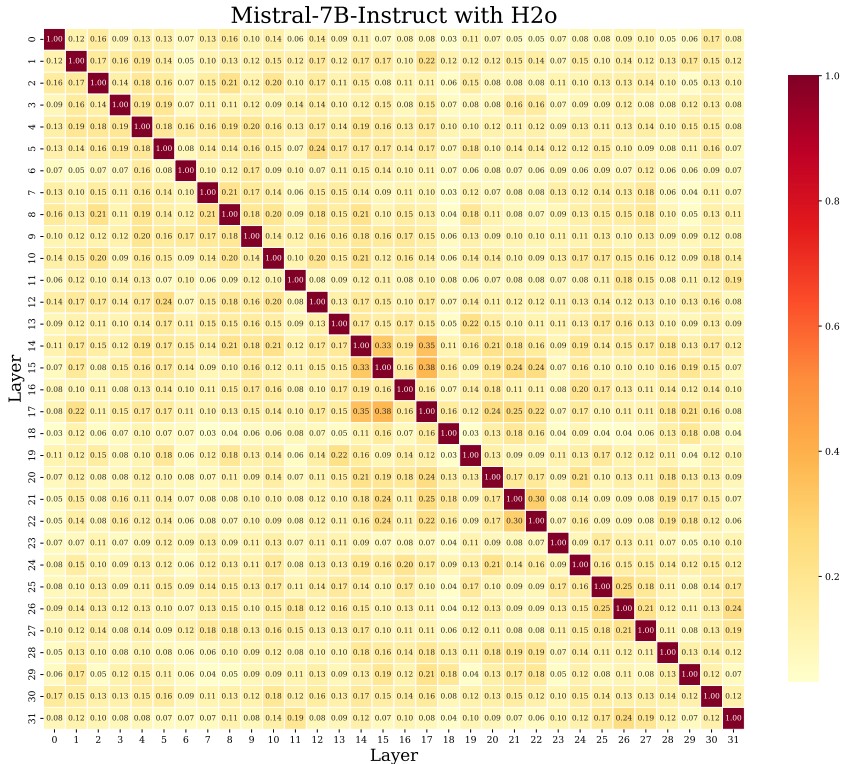

Figure 10: Layer-wise similarity heatmaps of the preserved KV cache indices by H2O on Mistral-7B-Instruct

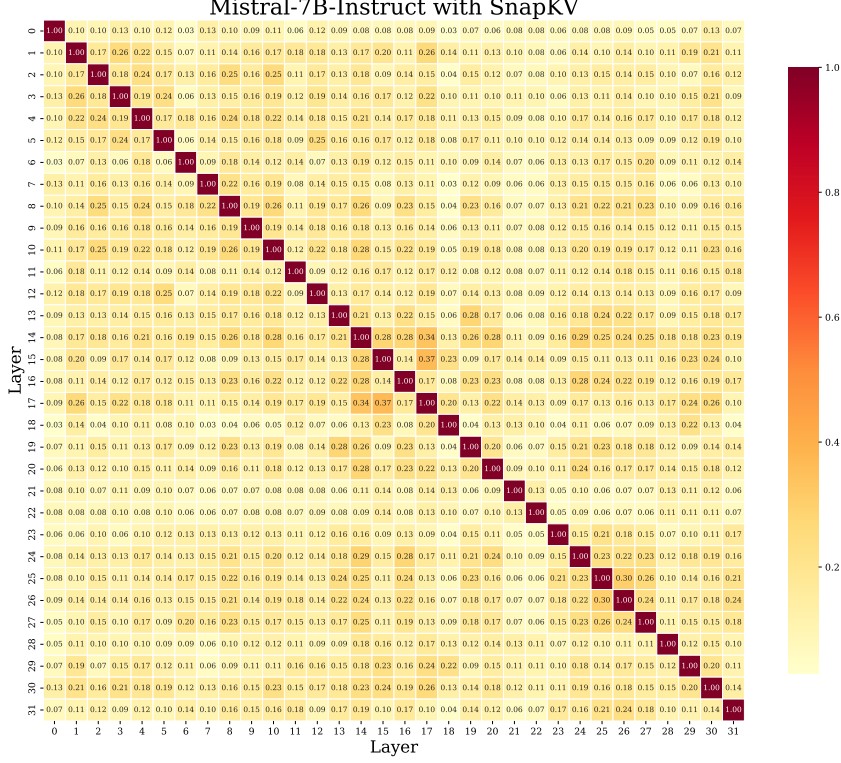

Figure 11: Layer-wise similarity heatmaps of the preserved KV cache indices by SnapKV on Mistral-7B-Instruct

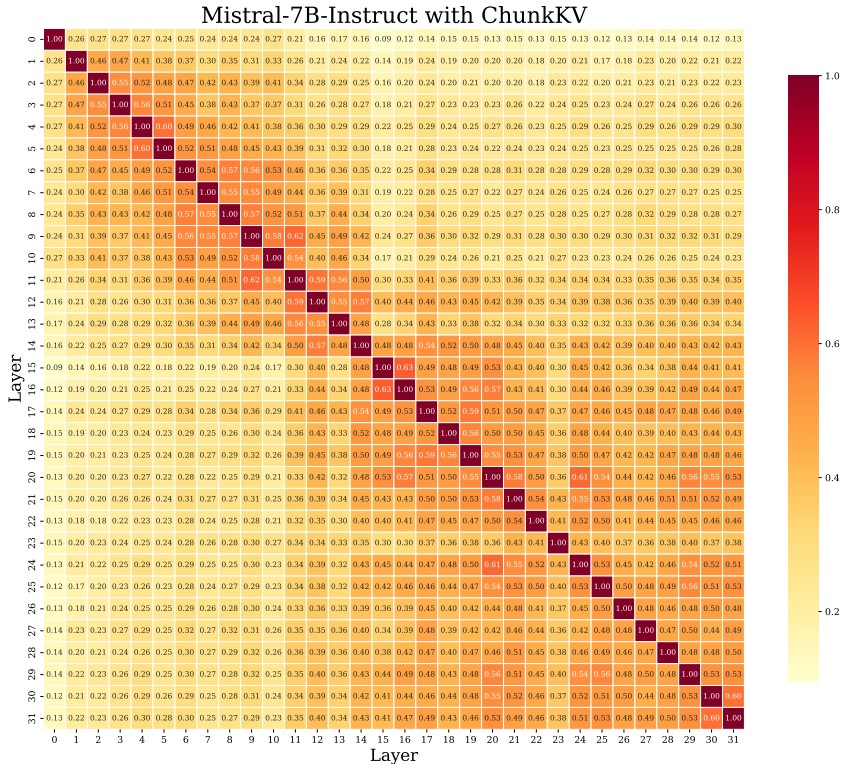

Figure 12: Layer-wise similarity heatmaps of the preserved KV cache indices by ChunkKV on Mistral-7B-Instruct

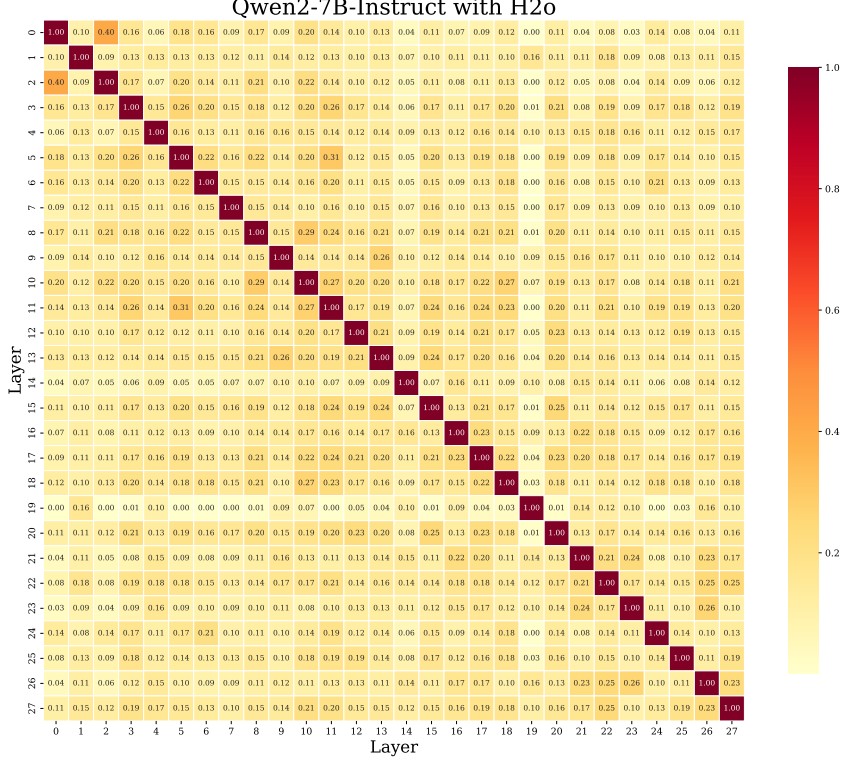

Figure 13: Layer-wise similarity heatmaps of the preserved KV cache indices by H2O on Qwen2-7B-Instruct

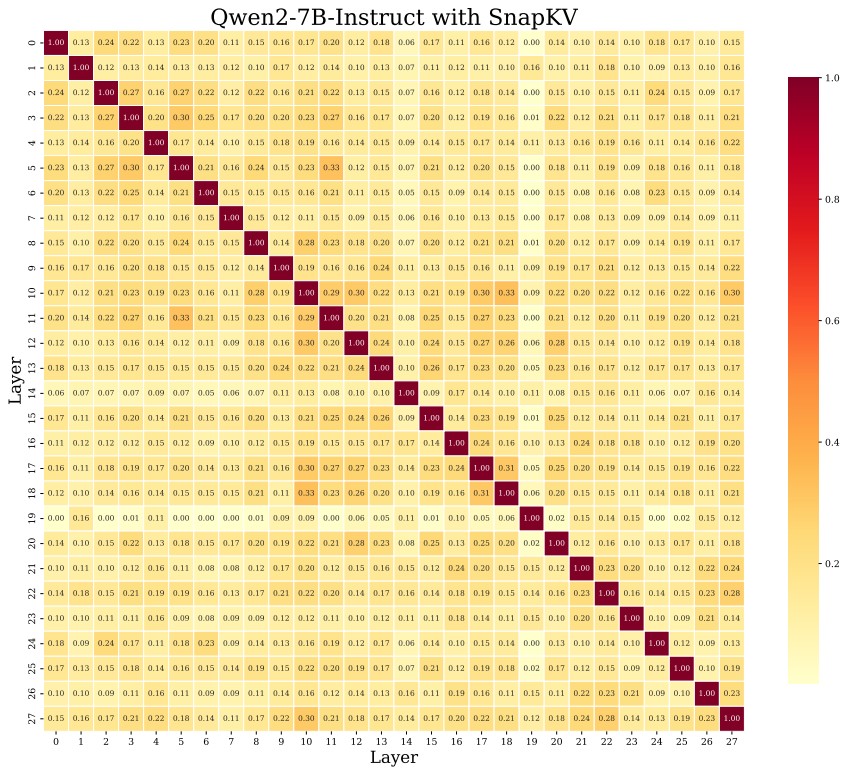

Figure 14: Layer-wise similarity heatmaps of the preserved KV cache indices by SnapKV on Qwen2-7B-Instruct

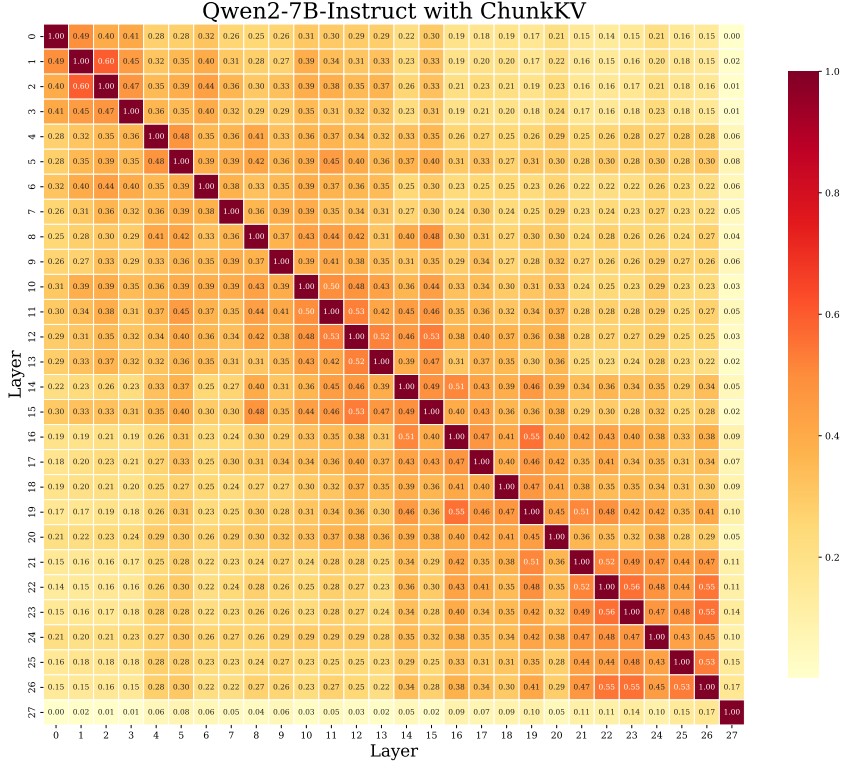

Figure 15: Layer-wise similarity heatmaps of the preserved KV cache indices by ChunkKV on Qwen2-7B-Instruct

Table 17: Comprehensive performance comparison of KV cache compression methods across LongBench English subtasks. Results are shown for various models and tasks, highlighting the effectiveness of different compression techniques.

| Method | Single-Document QA | | | Multi-Document QA | | | Summarization | | | Few-shot Learning | | | Synthetic | | Code | | Avg. ↑ |
|---|---|---|---|---|---|---|---|---|---|---|---|---|---|---|---|---|---|
| | NrtvQA | Qasper | MF-en | HotpotQA | 2WikiMQA | Musique | GovReport | QMSum | MultiNews | TREC | TriviaQA | SAMSum | PCount | PRe | Lcc | RB-P | |
| Avg len | 18,409 | 3,619 | 4,559 | 9,151 | 4,887 | 11,214 | 8,734 | 10,614 | 2,113 | 5,177 | 8,209 | 6,258 | 11,141 | 9,289 | 1,235 | 4,206 | |
| *LlaMa-3-8B-Instruct, KV Size = Full* | | | | | | | | | | | | | | | | | |
| FullKV | 25.70 | 29.75 | 41.12 | 45.55 | 35.87 | 22.35 | 25.63 | 23.03 | 26.21 | 73.00 | 90.56 | 41.88 | 4.67 | 69.25 | 58.05 | 50.77 | 41.46 |
| *LlaMa-3-8B-Instruct, KV Size Compression Ratio = 10%* | | | | | | | | | | | | | | | | | |
| StreamingLLM | 20.62 | 13.09 | 22.10 | 36.31 | 28.01 | 15.61 | 21.47 | 21.05 | 19.39 | 62.00 | 84.18 | 40.27 | 4.62 | 69.10 | 58.84 | 55.26 | 35.74 |
| H2O | 24.80 | 17.32 | 31.80 | 40.84 | 33.28 | 18.90 | 22.29 | 22.29 | 21.82 | 40.00 | 90.51 | 40.55 | 5.79 | **69.50** | 58.04 | 55.26 | 37.06 |
| SnapKV | 25.08 | 22.02 | **37.95** | 43.36 | 35.08 | 20.29 | 22.94 | 22.64 | 21.37 | 71.00 | 90.47 | 40.15 | 5.66 | 69.25 | 58.69 | 56.50 | 40.15 |
| PyramidKV | **25.58** | 20.77 | 35.85 | **43.80** | 33.03 | **21.45** | **23.68** | 22.26 | **21.85** | 71.50 | 90.47 | **41.66** | 5.84 | 69.25 | 58.52 | 55.91 | 40.08 |
| ChunkKV | 24.89 | **22.96** | 37.64 | 43.27 | **36.45** | 20.65 | 22.80 | **22.97** | 20.82 | **71.50** | **90.52** | 40.83 | **5.93** | 69.00 | **60.49** | **57.48** | **40.51** |
| *LlaMa-3-8B-Instruct, KV Size Compression Ratio = 20%* | | | | | | | | | | | | | | | | | |
| StreamingLLM | 23.35 | 18.97 | 32.94 | 42.39 | 29.37 | 18.76 | **25.78** | 21.92 | **25.16** | 71.00 | 88.85 | 40.82 | 5.04 | 69.00 | 56.46 | 51.12 | 38.80 |
| H2O | 25.60 | 21.88 | 35.36 | 42.06 | 32.68 | 19.72 | 23.54 | 22.77 | 22.72 | 45.50 | **90.57** | **41.67** | 5.51 | 69.25 | 54.97 | 50.95 | 37.79 |
| SnapKV | 25.50 | 25.95 | 38.43 | 44.12 | 35.38 | 20.49 | 24.85 | 23.36 | 23.51 | **72.50** | 90.52 | 40.91 | 5.23 | 69.25 | 56.74 | 51.75 | 40.53 |
| PyramidKV | 25.36 | 26.88 | 37.99 | 44.21 | **35.65** | **21.43** | 25.52 | **23.43** | 23.47 | 72.00 | 90.56 | 41.45 | 5.26 | **69.50** | 56.55 | 50.93 | 40.63 |
| ChunkKV | **26.13** | **28.43** | 38.59 | **44.46** | 34.13 | 21.06 | 24.72 | 23.11 | 22.91 | 71.50 | 90.56 | 41.51 | 5.09 | 69.00 | **58.17** | **52.51** | **40.74** |
| *LlaMa-3-8B-Instruct, KV Size Compression Ratio = 30%* | | | | | | | | | | | | | | | | | |
| StreamingLLM | 24.49 | 22.53 | 35.30 | **44.33** | 32.81 | 19.00 | **27.12** | 22.19 | **25.93** | 72.50 | 89.84 | 41.75 | **5.41** | 69.00 | 60.40 | 55.13 | 40.48 |
| H2O | 25.87 | 23.03 | 37.06 | 43.71 | 33.68 | 20.93 | 24.56 | 23.14 | 23.58 | 50.50 | **90.77** | 41.96 | 4.91 | 69.25 | 59.38 | 55.39 | 39.23 |
| SnapKV | 25.15 | 28.75 | **39.28** | 43.57 | 36.16 | 21.58 | 25.56 | **23.19** | 24.30 | **73.00** | 90.52 | 41.70 | 4.96 | 69.25 | 60.27 | 55.74 | 41.43 |
| PyramidKV | 25.42 | 27.91 | 38.81 | 44.15 | **36.28** | **21.72** | 26.50 | 23.10 | 24.28 | 72.00 | 90.56 | 41.87 | 4.67 | **69.50** | 60.09 | 55.19 | 41.37 |
| ChunkKV | **25.88** | **29.58** | 38.99 | 43.94 | 34.16 | 21.70 | 26.50 | 23.15 | 23.95 | 72.00 | 90.56 | **42.47** | 5.34 | 69.25 | **61.68** | **56.35** | **41.59** |
| *Mistral-7B-Instruct-v0.3, KV Size = Full* | | | | | | | | | | | | | | | | | |
| FullKV | 29.07 | 41.58 | 52.88 | 49.37 | 39.01 | 28.58 | 34.93 | 25.68 | 27.74 | 76.00 | 88.59 | 47.59 | 6.00 | 98.50 | 61.41 | 62.39 | 48.08 |
| *Mistral-7B-Instruct-v0.3, KV Size Compression Ratio = 10%* | | | | | | | | | | | | | | | | | |
| StreamingLLM | 25.15 | 25.47 | 30.08 | 44.39 | 32.49 | 19.40 | 24.11 | 20.85 | 19.55 | 65.00 | 88.21 | 44.83 | 4.50 | 79.50 | 59.48 | 58.82 | 40.11 |
| H2O | 29.35 | 33.39 | 50.39 | 49.58 | 36.76 | 27.42 | 25.16 | 24.75 | 22.12 | 42.00 | 89.00 | **47.04** | 5.50 | 98.50 | 57.58 | 59.24 | 43.61 |
| SnapKV | 28.54 | **36.88** | 53.42 | 50.15 | 38.17 | 27.99 | 26.67 | **25.21** | 22.33 | 72.00 | 89.36 | 45.44 | **5.50** | **99.00** | 59.79 | **61.63** | 46.38 |
| PyramidKV | 29.40 | 35.39 | 52.96 | 49.93 | 38.67 | 28.63 | **27.59** | 24.99 | 22.77 | 74.00 | **90.02** | 46.07 | 4.00 | 98.50 | 58.54 | 60.88 | 46.39 |
| ChunkKV | **29.75** | 36.82 | **53.99** | **50.33** | **38.72** | **29.01** | 27.03 | 24.76 | 21.42 | **76.00** | 88.73 | 46.49 | 5.00 | 98.00 | **59.98** | 61.47 | **46.71** |
| *Qwen2-7B-Instruct, KV Size = Full* | | | | | | | | | | | | | | | | | |
| FullKV | 25.11 | 42.64 | 44.29 | 14.25 | 13.22 | 9.08 | 36.38 | 23.43 | 26.53 | 77.00 | 89.99 | 44.88 | 6.75 | 75.92 | 60.17 | 61.84 | 40.71 |
| *Qwen2-7B-Instruct, KV Size Compression Ratio = 10%* | | | | | | | | | | | | | | | | | |
| StreamingLLM | 25.15 | 45.42 | 41.46 | 13.66 | 11.95 | 8.72 | 32.79 | 21.49 | **26.24** | 77.50 | 89.15 | 44.54 | 7.50 | 50.50 | 60.03 | 60.91 | 38.56 |
| H2O | 26.17 | 44.33 | 42.54 | 12.81 | 12.46 | 9.15 | **33.24** | 22.69 | 25.94 | 76.50 | 89.44 | 44.32 | 8.00 | 76.00 | **61.28** | **62.39** | 40.45 |
| SnapKV | 26.84 | **45.96** | **45.79** | 14.27 | **13.35** | 9.91 | 32.62 | 22.70 | 25.83 | 77.00 | 89.36 | 44.71 | 7.50 | 71.50 | 60.35 | 61.37 | 40.55 |
| PyramidKV | **27.51** | 44.45 | 43.59 | 13.35 | 13.13 | 9.12 | 32.28 | 22.60 | 25.45 | 77.00 | **89.44** | 44.53 | 7.00 | 73.50 | 60.91 | 61.24 | 40.31 |
| ChunkKV | 26.48 | 44.19 | 45.04 | **15.94** | 12.60 | **10.52** | 32.38 | **22.87** | 25.91 | **77.50** | 89.22 | 44.78 | **8.50** | **76.50** | 60.64 | 61.32 | **40.88** |

Table 18: Comprehensive performance comparison of ChunkKV across LongBench English subtasks with 70B model.

| Method | Single-Document QA | | | Multi-Document QA | | | Summarization | | | Few-shot Learning | | | Synthetic | | Code | | Avg. ↑ |
|---|---|---|---|---|---|---|---|---|---|---|---|---|---|---|---|---|---|
| | NrtvQA | Qasper | MF-en | HotpotQA | 2WikiMQA | Musique | GovReport | QMSum | MultiNews | TREC | TriviaQA | SAMSum | PCount | PRe | Lcc | RB-P | |
| Avg len | 18,409 | 3,619 | 4,559 | 9,151 | 4,887 | 11,214 | 8,734 | 10,614 | 2,113 | 5,177 | 8,209 | 6,258 | 11,141 | 9,289 | 1,235 | 4,206 | |
| *LlaMa-3-70B-Instruct, KV Size = Full* | | | | | | | | | | | | | | | | | |
| FullKV | 27.56 | 46.00 | 49.75 | 50.86 | 56.16 | 29.77 | 31.66 | 21.80 | 27.42 | 75.50 | 92.58 | 46.08 | 12.00 | 73.50 | 45.02 | 68.74 | 47.15 |
| *LlaMa-3-70B-Instruct, KV Cache Size = 256* | | | | | | | | | | | | | | | | | |
| H2O | 26.42 | 37.52 | 36.23 | 47.82 | 49.34 | 24.30 | 26.96 | 20.67 | **26.50** | 72.00 | 92.16 | 43.56 | 12.00 | 73.00 | 47.21 | 63.13 | 43.67 |
| SnapKV | 26.79 | 42.23 | 48.78 | **50.89** | 55.46 | 28.14 | 23.85 | 22.20 | 24.62 | 73.00 | 93.05 | 44.53 | 12.00 | 72.50 | **54.24** | 65.60 | 46.11 |
| ChunkKV | **27.37** | **45.95** | 49.23 | 50.51 | 55.49 | 30.28 | 30.47 | 22.63 | 26.30 | 75.50 | 92.58 | 46.51 | 12.00 | 73.50 | 49.51 | **68.05** | **47.24** |

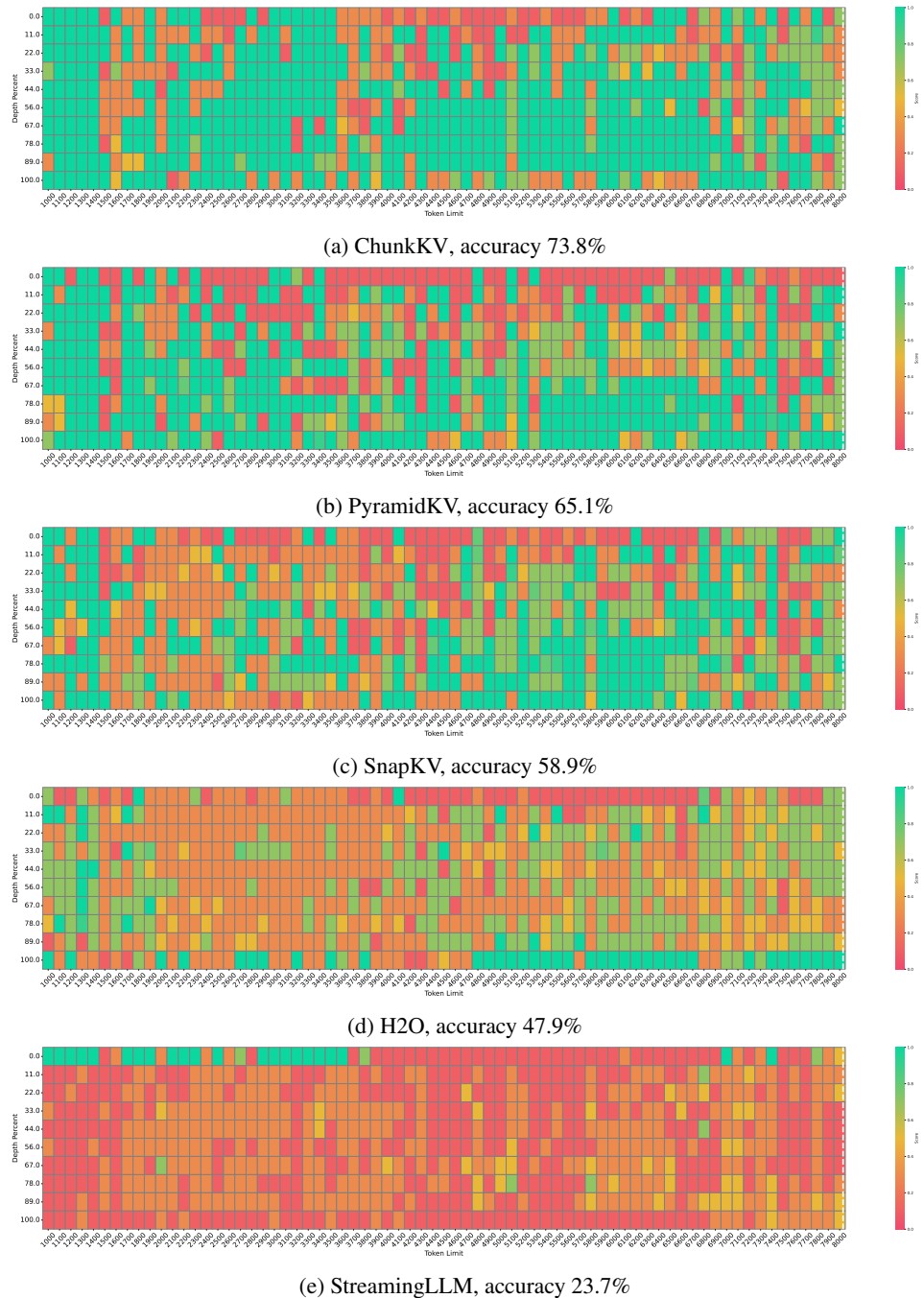

(a) ChunkKV, accuracy 73.8%

(b) PyramidKV, accuracy 65.1%

(c) SnapKV, accuracy 58.9%

(d) H2O, accuracy 47.9%

(e) StreamingLLM, accuracy 23.7%

Figure 16: NIAH benchmark for LLaMA-3-8B-Instruct with KV cache size=128 under 8k context length

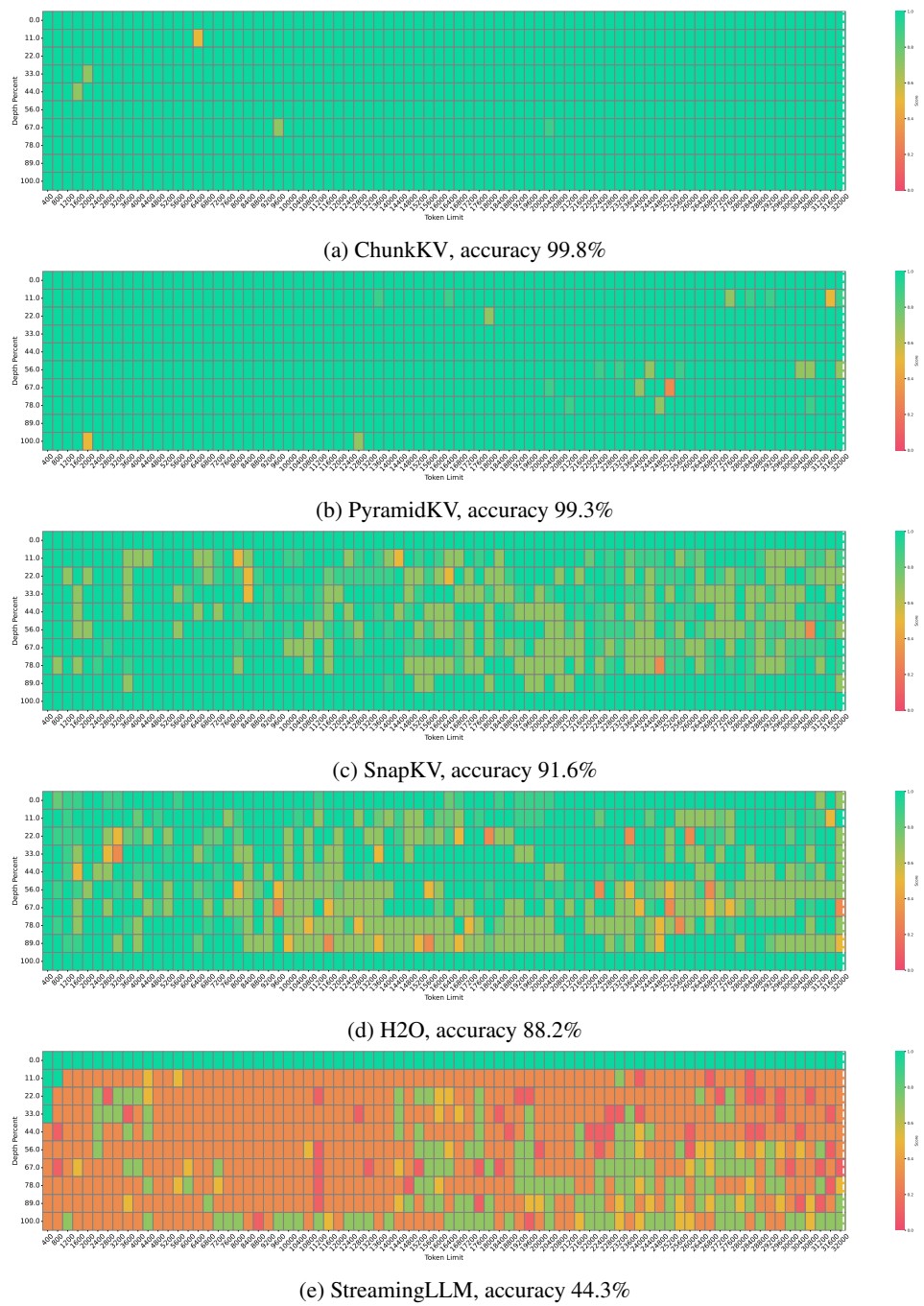

(a) ChunkKV, accuracy 99.8%

(b) PyramidKV, accuracy 99.3%

(c) SnapKV, accuracy 91.6%

(d) H2O, accuracy 88.2%

(e) StreamingLLM, accuracy 44.3%

Figure 17: NIAH benchmark for Mistral-7B-Instruct with KV cache size=128 under 32k context length

## B.4  Chunk Size

Table 19 and 20 show the performance of ChunkKV with different comperession ratios and different chunk sizes on the LongBench and NIAH. We conducted extensive experiments across different compression ratios and KV cache sizes to shows the effectiveness of ChunkKV and the chunk size is robust.

Table 19: LongBench Performance with Different Chunk Sizes and Compression Ratios for LLaMA-3-8B-Instruct

| Compression | Chunk Size | | | | | | |
|---|---|---|---|---|---|---|---|
| Rate | 1 | 3 | 5 | 10 | 15 | 20 | 30 |
| 10% | 37.32 | 40.49 | 40.47 | **40.51** | 40.21 | 40.05 | 39.57 |
| 20% | 38.80 | 40.66 | 40.57 | **40.74** | 40.53 | 40.46 | 40.04 |
| 30% | 39.23 | 41.02 | 41.29 | **41.59** | 41.38 | 41.33 | 41.02 |

Table 20: NIAH Performance with Different Chunk Sizes and KV Cache Sizes for LLaMA-3-8B-Instruct

| KV Cache | Chunk Size | | | | | | |
|---|---|---|---|---|---|---|---|
| Size | 1 | 3 | 5 | 10 | 15 | 20 | 30 |
| 96 | 41.0 | 63.2 | 65.2 | **70.3** | 67.2 | 65.3 | 53.1 |
| 128 | 47.9 | 65.6 | 69.1 | **73.8** | 72.3 | 72.0 | 71.2 |
| 256 | 61.7 | 70.3 | 71.2 | **74.1** | 73.2 | 72.3 | 71.1 |
| 512 | 68.6 | 72.6 | 72.5 | **74.5** | 74.3 | 74.0 | 72.6 |

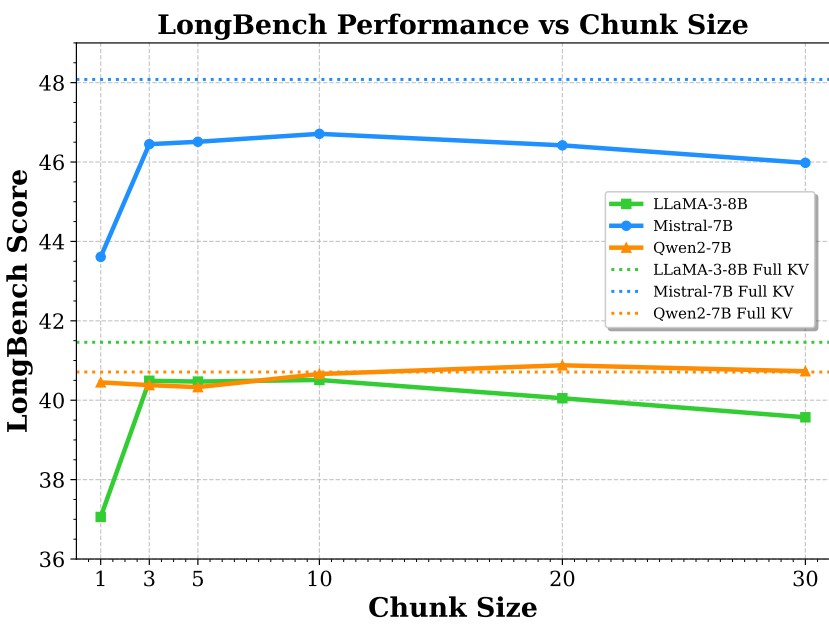

Figure 18: LongBench Performance Comparison with different chunk size under 10% compression rate.

Figure 18 shows the performance of the ChunkKV with different chunk size on the LongBench and NIAH benchmarks. The three colorful curves represent three LLMs with different chunk sizes, and the colorful dashed line is the corresponding FullKV performance. From Figure 18, we can observe that the LongBench performance of ChunkKV is not significantly affected by the chunk size, with

performance variations less than 1%. The three curves are closely aligned, indicating that chunk sizes in the range of $\{10, 20\}$ exhibit better performance.

Table 21 and Figure 19 shows that the ChunkKV with different chunk sizes on GSM8K displays the same curve pattern as LongBench. The CoT prompt length for GSM8K is only 1K tokens, so the optimal chunk size range is smaller.

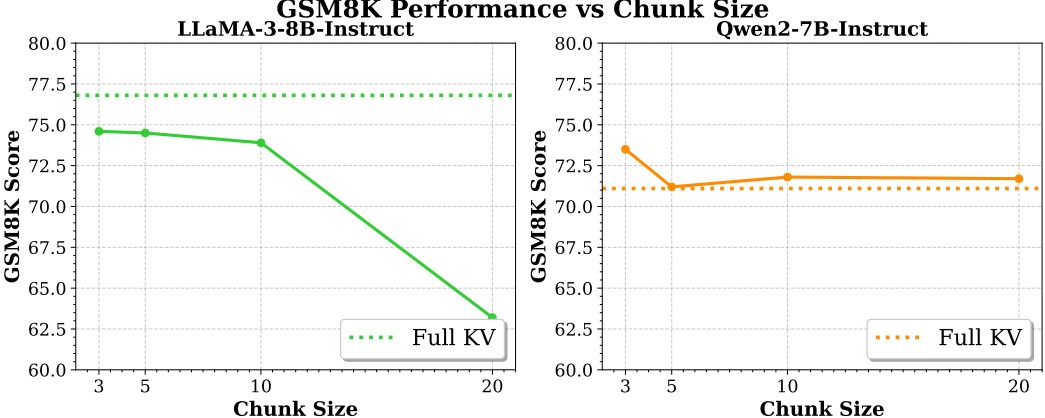

Figure 19: GSM8K Performance Comparison with different chunk size

Table 21: GSM8K Performance Comparison with different chunk sizes

| Model | Chunk Size | | | | Full KV |
|---|---|---|---|---|---|
| | 3 | 5 | 10 | 20 | |
| LLaMA-3-8B-Instruct | **74.6** | 74.5 | 73.9 | 63.2 | 76.8 |
| Qwen2-7B-Instruct | **73.5** | 71.2 | 71.8 | 71.7 | 71.1 |

## B.5   Multi-Lingual

Table 22 is the Chinese support model Qwen2-7B-Instruct evaluated on the LongBench Chinese subtask, where ChunkKV achieves better performance than other compression methods and the full KV cache performance. Both the English and Chinese results indicate that ChunkKV is a promising approach for maintaining crucial information in the KV cache.

Table 22: Performance comparison of Chinese subtask on LongBench for Qwen2-7B-Instruct.

| Method | Single-Document QA | Multi-Document QA | Summarization | Few-shot Learning | Synthetic | |
|---|---|---|---|---|---|---|
| | MF-zh | DuReader | VCSum | LSHT | PR-zh | **Avg. ↑** |
| Avg len | 6,701 | 15,768 | 15,380 | 22,337 | 6,745 | |
| Qwen2-7B-Instruct, KV Size = Full | | | | | | |
| FullKV | 39.17 | 23.63 | 16.21 | 43.50 | 70.50 | 38.60 |
| Qwen2-7B-Instruct, KV Size Compression Ratio = 10% | | | | | | |
| StreamingLLM | 38.05 | **23.24** | 15.92 | 40.50 | 44.50 | 32.44 |
| H2O | 37.99 | 19.58 | 16.16 | 41.67 | 67.35 | 36.55 |
| SnapKV | 44.25 | 20.27 | 16.24 | **44.50** | 68.10 | 38.67 |
| PyramidKV | 36.57 | 20.56 | 16.15 | 43.50 | 66.50 | 36.55 |
| **ChunkKV** | **45.92** | 20.15 | **16.37** | 43.75 | **71.10** | **39.45** |

## B.6   KV Cache Quantization

For comprehensively evaluate the effectiveness of ChunkKV, we conducted experiments comparing ChunkKV with quantization methods KIVI [45]. While both approaches aim to optimize LLM

inference, they operate on fundamentally different principles: quantization reduces KV matrix precision, whereas our eviction method reduces KV matrix size.

From an implementation perspective, quantization methods require the full KV cache during prefilling to produce quantized representations, which are then used during decoding. In contrast, ChunkKV employs token removal prior to prefilling, enabling operation with a compressed cache throughout the entire inference process. This distinction creates different efficiency profiles, and each method offers unique advantages.

The following tables present our detailed analysis comparing ChunkKV with KIVI, across both performance and efficiency metrics on the LLaMA-3-8B-Instruct model. It should be noted that due to KIVI's dependency on older Python versions, there may be some discrepancies between the efficiency results reported here and those in Table 8 of our paper.

Table 23: Comprehensive performance comparison of ChunkKV and quantization methods across LongBench English subtasks.

| Method | Single-Document QA | | | Multi-Document QA | | | Summarization | | | Few-shot Learning | | | Synthetic | | Code | | Avg. ↑ |
|---|---|---|---|---|---|---|---|---|---|---|---|---|---|---|---|---|---|
| | NrtvQA | Qasper | MF-en | HotpotQA | 2WikiMQA | Musique | GovReport | QMSum | MultiNews | TREC | TriviaQA | SAMSum | PCount | PRe | Lcc | RB-P | |
| Avg len | 18,409 | 3,619 | 4,559 | 9,151 | 4,887 | 11,214 | 8,734 | 10,614 | 2,113 | 5,177 | 8,209 | 6,258 | 11,141 | 9,289 | 1,235 | 4,206 | |
| LlaMa-3-8B-Instruct, KV Size = Full | | | | | | | | | | | | | | | | | |
| FullKV | 25.70 | 29.75 | 41.12 | 45.55 | 35.87 | 22.35 | 25.63 | 23.03 | 26.21 | 73.00 | 90.56 | 41.88 | 4.67 | 69.25 | 58.05 | 50.77 | 41.46 |
| ChunkKV with different compression ratios | | | | | | | | | | | | | | | | | |
| **ChunkKV-10%** | 24.89 | 22.96 | 37.64 | 43.27 | 36.45 | 20.65 | 22.80 | 22.97 | 20.82 | 71.50 | 90.52 | 40.83 | 5.93 | 69.00 | 60.49 | 57.48 | 40.51 |
| **ChunkKV-20%** | 26.13 | 28.43 | 38.59 | 44.46 | 34.13 | 21.06 | 24.72 | 23.11 | 22.91 | 71.50 | 90.56 | 41.51 | 5.09 | 69.00 | 58.17 | 52.51 | 40.74 |
| **ChunkKV-30%** | 25.88 | 29.58 | 38.99 | 43.94 | 34.16 | 21.70 | 26.50 | 23.15 | 23.95 | 72.00 | 90.56 | 42.47 | 5.34 | 69.25 | 61.68 | 56.35 | **41.59** |
| KIVI with different quantization nbits | | | | | | | | | | | | | | | | | |
| **KIVI-2bits** | 25.45 | 29.45 | 40.85 | 45.25 | 35.55 | 22.05 | 25.48 | 22.95 | 26.06 | 70.25 | 89.00 | 41.73 | 4.25 | 64.00 | 59.72 | 46.70 | 40.54 |
| **KIVI-4bits** | 25.57 | 29.60 | 41.00 | 45.40 | 35.70 | 22.20 | 28.78 | 23.00 | 26.49 | 72.50 | 89.84 | 42.16 | 4.35 | 64.52 | 59.91 | 46.84 | 41.11 |
| **KIVI-8bits** | 25.70 | 29.73 | 41.11 | 45.52 | 35.81 | 22.34 | 25.62 | 23.01 | 26.19 | 72.97 | 90.54 | 41.83 | 4.62 | 69.22 | 58.02 | 50.74 | 41.43 |

Table 23 demonstrates that ChunkKV with 30% compression ratio achieves comparable performance (41.59) to KIVI with 8-bit quantization (41.43) across LongBench subtasks. Notably, ChunkKV shows particularly strong results in code-related tasks, while KIVI maintains better performance in single-document QA scenarios.

These results highlight the complementary strengths of eviction and quantization approaches. ChunkKV's semantic chunk-based method delivers particularly strong efficiency benefits while maintaining competitive performance with state-of-the-art quantization techniques.

## B.7 Comparison with Orthogonal and Training-Based Methods

To better position ChunkKV in the broader landscape of inference optimization, we conducted additional experiments comparing our training-free eviction method against Palu [46], a recent training-based compression method. The experiments were conducted on the LongBench benchmark with the LLaMA-3-8B-Instruct model.

Table 24: Performance comparison of ChunkKV with Palu (training-based) on the LongBench benchmark (LLaMA-3-8B-Instruct). ChunkKV demonstrates a superior performance-efficiency trade-off compared to training-based methods on this diverse benchmark and remains competitive with orthogonal quantization techniques.

| Method | Single-Doc QA | Multi-Doc QA | Summarization | Few-shot | Synthetic | Code | Average Score ↑ |
|---|---|---|---|---|---|---|---|
| FullKV (FP16) | 32.19 | 34.59 | 24.96 | 68.48 | 36.96 | 54.41 | 41.46 |
| *Eviction Methods (Ours)* | | | | | | | |
| ChunkKV (10% Ratio) | 28.50 | 33.46 | 22.20 | 67.62 | 37.47 | 58.98 | 40.51 |
| ChunkKV (30% Ratio) | 31.48 | 33.27 | 24.53 | 68.34 | 37.30 | 59.02 | 41.59 |
| *Training-Based Methods* | | | | | | | |
| Palu (50% Ratio) | 8.77 | 7.43 | 20.71 | 61.62 | 8.59 | 18.43 | 22.48 |
| Palu (70% Ratio) | 10.06 | 8.31 | 23.71 | 68.64 | 35.00 | 38.97 | 28.84 |

The results in Table 24 highlight key differences. Palu, as a training-based method, experiences significant performance degradation on the diverse LongBench benchmark, which represents an Out-Of-Distribution (OOD) challenge relative to its training data. In contrast, our training-free method, ChunkKV, demonstrates robust performance.

## B.8 Efficiency results

Table 25 shows the efficiency results for different KV cache strategies with varying output lengths and the metrics are Time to First Token (TTFT) and Token Processing Time (TPOT).

Table 25: Efficiency Results for Different KV Cache Strategies with Varying Output Lengths

| Method | Input | Output | TTFT(s) ↓ | TPOT(ms/token) ↓ |
|---|---|---|---|---|
| **Output Length = 1024** | | | | |
| FullKV | 4096 | 1024 | 1.17 | 42.58 |
| StreamingLLM | 4096 | 1024 | 1.02 (12.8%) | 35.94 (15.6%) |
| SnapKV | 4096 | 1024 | 1.04 (11.1%) | 36.24 (14.9%) |
| ChunkKV | 4096 | 1024 | 1.03 (12.0%) | 36.64 (13.9%) |
| ChunkKV_reuse | 4096 | 1024 | 1.00 (14.5%) | 36.47 (14.3%) |
| **Output Length = 4096** | | | | |
| FullKV | 4096 | 4096 | 1.17 | 42.85 |
| StreamingLLM | 4096 | 4096 | 1.09 (6.8%) | 40.01 (6.6%) |
| SnapKV | 4096 | 4096 | 1.10 (6.0%) | 40.24 (6.1%) |
| ChunkKV | 4096 | 4096 | 1.10 (6.0%) | 40.17 (6.3%) |
| ChunkKV_reuse | 4096 | 4096 | 1.08 (7.7%) | 40.16 (6.3%) |

Table 26: Latency and throughput comparison on LLaMA-3-8B-Instruct, including long-context scenarios up to 16k tokens. Percentages in parentheses indicate improvements over the FullKV baseline. ChunkKV with layer-wise reuse (ChunkKV_reuse) consistently delivers the best efficiency gains, especially in long-context settings.

| Method | Sequence Length | | Performance Metrics | |
|---|---|---|---|---|
| | Input | Output | Latency(s) ↓ | Throughput(T/S) ↑ |
| *4k Context Length* | | | | |
| FullKV | 4096 | 1024 | 43.60 | 105.92 |
| SnapKV | 4096 | 1024 | 37.92 (13.0%) | 120.42 (13.7%) |
| ChunkKV | 4096 | 1024 | 37.52 (13.9%) | 118.85 (12.2%) |
| ChunkKV_reuse | 4096 | 1024 | **37.35 (14.3%)** | **124.09 (17.2%)** |
| *16k Context Length* | | | | |
| FullKV | 16384 | 1024 | 49.60 | 323.24 |
| SnapKV | 16384 | 1024 | 39.20 (21.0%) | 381.11 (17.9%) |
| ChunkKV | 16384 | 1024 | 38.82 (21.7%) | 381.61 (18.1%) |
| ChunkKV_reuse | 16384 | 1024 | **36.96 (25.5%)** | **389.21 (20.4%)** |

## C   Theoretical Understanding

In this section, we provide the theoretical interpretation from the perspective from the In-context learning (ICL) to further understand how ChunkKV outperforms token-level KV cache compression.

**Pretraining Data Distribution.** Given a set of concepts $\Theta$ and a concept $\theta \in \Theta$, we define the pretraining data is sampled from $p(o_1, \ldots, o_T) = \int_{\theta \in \Theta} p(o_1, \ldots, o_T | \theta) p(\theta) d\theta$ [24]. Each token $o$ is sampled from a vocabulary $\mathbb{O}$. For simplicity, we write $o_{1:t} = o_1 \ldots o_t$.

**Language Modeling.** Current LLMs [1, 5, 24] usually utilize the next word prediction as the language modelling, which predicts the next token $o_t$ given the previous tokens $o_1 \ldots o_{t-1}$ for all $t = 1, \ldots, T$. Formally, a language modelling can be writen as the distribution $f(o_t | o_{1:t-1})$. And it is pretrained on

a huge corpus sampled from the pretraining distribution $p(o_1, \ldots, o_{t+1})$ [24]. Considering the large scale of the model size and dataset size, it can be assumed that the $f(o_1 \ldots o_{t+1})$ has been aligned with the $p(o_1 \ldots o_{t+1})$ [24].

**Prompt Distribution.** Following [24], a prompt is composed of an input token sequence $x$ followed by an output token $y$. Then, the $i$-th training example [†] that can appear in any place in the whole prompt $o_{1:T}$ is defined as $O_i$ consisting of an input $x_i = O_i[1:k-1]$ (the first $k-1$ tokens) followed by the output $y_i = O_i[k]$ at the end, where the length $k$ is fixed for simplicity.

The $i$-th training example is independently generated as follows: 1) Generate a start hidden state $h_i^{\text{start}}$ from a *prompt start distribution* $p_{\text{prompt}}$; 2) Given $h_i^{\text{start}}$, generate the example sequence $O_i = [x_i, y_i]$ from $p(O_i|h_i^{\text{start}}, \theta^\star)$. The test input $x_{\text{test}} = x_{n+1}$ is sampled similarly. Then, the prompt consists of a sequence of training examples $(S_n)$ followed by the example $x_{\text{test}}$:

$$[S_n, x_{\text{test}}] = [x_1, y_1, x_2, y_2, \ldots, x_n, y_n, x_{\text{test}}] \sim p_{\text{prompt}}. \tag{2}$$

**In-context learning setups and Assumptions.** We follow other settings and assumptions in [24]. With the greedy decoding [47], sampling the next token from the language modeling $f(o_t|o_{1:t-1})$ becomes the predictor as $y = \arg\max_{o_t} f(o_t|o_{1:t-1})$.

Thus, for $[S_n, x_{\text{test}}]$, the in-context learning predictor can be written as $f_n(x_{\text{test}}) := \arg\max_y p(y|S_n, x_{\text{test}})$, which outputs the most likely prediction over the *pretraining distribution* conditioned on the *prompt distribution*. Its expected 0-1 error with $n$ examples is $L_{\text{0-1}}(f_n) = \mathbb{E}_{x_{\text{test}}, y_{\text{test}} \sim p_{\text{prompt}}}[\mathbf{1}[f_n(x_{\text{test}}) \neq y_{\text{test}}]]$.

We define $p_\theta^i(o) := p(O[i] = o|O[1:i-1], \theta)$ of the $i$-th token with previous tokens and the analogous distribution $p_{prompt}^i := p_{prompt}(O[i] = o|O[1:i-1])$ under the prompt distribution. Following [24], there is a distinguishability condition formalizes when in-context learning occurs giving the concept $\theta$.

The distinguishability condition is dependent on a KL divergence between the previous two distributions and the error terms $\epsilon_\theta$ resulting from the distribution mismatch between the prompt and the pertaining distributions for each example. Letting $p_\theta^i(o)$ and $p_{prompt}^i$ correspond to the concept $\theta$ and $\theta^\star$.

**Condition 1** (distinguishability [24]). *The $\theta^\star$ is distinguishable if for all $\theta \in \Omega$, $\theta \neq \theta^\star$,*

$$\sum_{i=1}^k KL_i(\theta^\star||\theta) > \epsilon_\theta, \tag{3}$$

*where the $KL_i(\theta^\star||\theta) := \mathbb{E}_{O[1:i-1] \sim p_{prompt}}[KL(p_{prompt}^i||p_\theta^i)]$.*

**Noises from KV Cache Compression.** Naturally, because of the sparsified KV cache, some history tokens in $o_{1:t-1}$ at different layers lost its attention score calculation with respect to the next word prediction $o_t$. We can regard this as the noise added onto the $o_{1:t-1}$. Thus, distincting $\theta^\star$ from $\theta$ requires larger KL divergence. Following [48], we provide the following second condition about the distinguishability with the KV cache sparsity.

**Condition 2** (distinguishability under sparsified KV cache). *With the noise introduced by the sparsified KV cache of the sparse ratio $r$, the distribution mismatch between the prompt and the pretraining distribution that is approximated by LLM is enlarged, resulting in a varied requirement with error term $\xi_\theta(r)$ for $\theta^*$ being distinguishable if for all $\theta \in \Theta$, $\theta \neq \theta^*$,*

$$\sum_{i=1}^k KL_i(\theta^*||\theta) > \epsilon_\theta + \xi_\theta(r), \quad \text{where} \quad \xi_\theta(r) \propto r. \tag{4}$$

**Lemma 1** (noisy-relaxed bound in [24, 48]). *let $\mathcal{B}$ denotes the set of $\theta$ which does not satisfy Condition 1. We assume that $KL(p_{prompt}(y_{test}|x_{test})||p(y_{test}|x_{test}, \theta)$ is bounded for all $\theta$ and that $\theta^\star$ minimizes the multi-class logistic risk as,*

$$L_{CE}(\theta) = -\mathbb{E}_{x_{test} \sim p_{prompt}}[p_{prompt}(y_{test}|x_{test}) \cdot \log p(y_{test}|x_{test}, \theta)]. \tag{5}$$

---

[†]Here, training example in prompts means happens during the prompt learning.

*If*

$$\mathbb{E}_{x_{test} \sim p_{prompt}}[KL(p_{prompt}(y_{test}|x_{test})||p(y_{test}|x_{test}, \theta))] \leq (\epsilon_\theta + \xi_\theta(r)), \quad \forall \quad \theta \in \mathcal{B}, \quad (6)$$

*then*

$$\lim_{n \to \infty} L_{0-1}(f_n) \leq \inf_f L_{0-1}(f) + g^{-1}\left(\sup_{\theta \in \mathcal{B}}(\epsilon_\theta)\right), \quad (7)$$

*where $g(\nu) = \frac{1}{2}\big((1-\nu)\log(1-\nu) + (1+\nu)\log(1+\nu)\big)$ is the calibration function [49, 50] for the multiclass logistic loss for $\nu \in [0,1]$.*

Following [24, 51], KL divergence is assumed to haver the 2nd-order Taylor expansion with the concept $\theta$. Then, we have the following theorem and proof.

**Theorem 1.** *[24, 48] Let the set of $\theta$ which does not satisfy Equation 3 in Condition 1 to be $\mathcal{B}$. Assume that KL divergences have a 2nd-order Taylor expansion around $\theta^\star$:*

$$\forall j > 1, \quad KL_i(\theta^\star||\theta) = \frac{1}{2}(\theta - \theta^\star)^\top I_{j,\theta^*}(\theta - \theta^\star) + O(\|\theta - \theta^\star\|^3) \quad (8)$$

*where $I_{j,\theta^*}$ is the Fisher information matrix of the $j$-th token distribution with respect to $\theta^\star$. Let $\gamma_{\theta^*} = \frac{\max_j \lambda_{max}(I_{j,\theta^*})}{\min j \lambda_{min}(I_{j,\theta^*})}$ where $\lambda_{max}, \lambda_{min}$ return the largest and smallest eigenvalues. Then for $k \geq 2$ and as $n \to \infty$, the 0-1 risk of the in-context learning predictor $f_n$ is bounded as*

$$\lim_{n \to \infty} L_{0\text{-}1}(f_n) \leq \inf_f L_{0\text{-}1}(f) + g^{-1}\left(O\left(\frac{\gamma_{\theta^*} \sup_{\theta \in \mathbb{B}}(\epsilon_\theta + \xi_\theta(r))}{k-1}\right)\right) \quad (9)$$

*Proof.* [24] By the Taylor expansion on $\theta$, we have for any $\theta$ in $\mathcal{B}$ that

$$\sum_{j=2}^{k} KL_i(\theta^\star||\theta) \geq \frac{1}{2}\sum_{j=2}^{k}(\theta - \theta^\star)^\top I_{j,\theta^*}(\theta - \theta^\star) + (k-1)O(\|\theta - \theta^\star\|^3) \quad (10)$$

$$\geq \frac{1}{2}(k-1)\lambda_{\min}(I_{j,\theta^*})\|\theta - \theta^\star\|^2 \quad (11)$$

$$\implies \|\theta - \theta^\star\|^2 \leq \frac{(\epsilon_\theta + \xi_\theta(r))}{\frac{1}{2}(k-1)(\min_j \lambda_{\min}(I_{j,\theta^*}))}. \quad (12)$$

We can bound the last KL term ($k$-th token) with the above term:

$$KL_k(\theta^\star||\theta) = \frac{1}{2}(\theta - \theta^\star)^\top I_{k,\theta^*}(\theta - \theta^\star) + O(\|\theta - \theta^\star\|^3) \quad (13)$$

$$\leq \frac{1}{2}(\max_j \lambda_{\max}(I_{j,\theta^*}))\|\theta - \theta^\star\|^2 + O(\|\theta - \theta^\star\|^2) \quad (14)$$

$$\leq \frac{(\epsilon_\theta + \xi_\theta(r))(\max_j \lambda_{\max}(I_{j,\theta^*}) + O(1))}{(k-1)\min_j \lambda_{\min}(I_{j,\theta^*})}. \quad (15)$$

Rearranging above equation, and with $KL_k(\theta^\star||\theta) = \mathbb{E}_{x_{test} \sim p_{prompt}}[KL(p_{prompt}(y_{test}|x_{test})||p(y_{test}|x_{test}, \theta))]$, there is

$$\mathbb{E}_{x_{test} \sim p_{prompt}}[KL(p_{prompt}(y_{test}|x_{test})||p(y_{test}|x_{test}, \theta))] \leq \frac{(\epsilon_\theta + \xi_\theta(r))(\max_j \lambda_{\max}(I_{j,\theta^*}) + O(1))}{(k-1)\min_j \lambda_{\min}(I_{j,\theta^*})} \quad (16)$$

Combining Equation 16 with Equation 6 into Lemma 1 completes the proof. $\square$

**KV Cache Sparsification.** Revisiting the Equation 4 in Condition 2, the $\xi_\theta(r)$ is enlarged with the sparsity ratio $r$. The higher compression ratio $r$ (means that more KV cache are discarded), the more noise $\xi_\theta(r)$. Then it leads to the higher bound of the $\lim_{n\to\infty} L_{0-1}(f_n)$ in Equation 5 in Lemma 1. Next, we discuss how KV cache compression influences the Equation 4.

**Token-level Importance Measure.** The token-level KV cache methods usually calculate the importance of different tokens. Then, the KV cache with indexes that have higher importance will

be preserved. Such indexes are normaly choosed as the attention score. Considering that the case in Figure 1, where each token in the $i$-th training $^\dagger$ example sequence ($O_i = [x_i, y_i]$) might be compressed, and tokens are sparsified concretely without out dependency to other tokens. However, in the generation process of the $i$-th training example, $O_i = [x_i, y_i]$ is sampled from $p(O_i|h_i^{\text{start}}, \theta^\star)$ and $p_\theta^j(o) := p(O[j] = o|O[1:j-1], \theta)$ of the $j$-th token with previous tokens and the analogous distribution $p_{prompt}^j := p_{prompt}(O[j] = o|O[1:j-1])$. And the KL divergence is defined as $\text{KL}_j(\theta^\star||\theta) := \mathbb{E}_{O[1:j-1] \sim p_{prompt}}[\text{KL}(p_{prompt}^j || p_\theta^j)]$, which means that in a training example $O_i = [x_i, y_i] = O_i[1:k]$, each token $O_i[j]$ has strong dependency with $O_i[1:j-1]$, noises on previous any $j$-th token will influence the distinguishability on the following tokens, i.e. requiring larger $\{\text{KL}_u(\theta^\star||\theta)\}_{u>j}$.

On the other hand, the token-level sparsity enlarges the requirement on the distinguishability uniformly for each example $O_i$ (the case in Figure 1), which uniformly loses the bound of $L_{0\text{-}1}(f_n)$ as in Equation 9.

**Chunk-level Importance Measure.** Different from token-level importance measure, ChunkKV regards tokens in a continuous window as a basic unit that should be left or discarded as a whole. The preserved window can be regarded as saving the complete $O_i = [x_i, y_i]$ without noise. Thus, ChunkKV reduces the noise $\xi_\theta(r)$ for the preserved $O_i$, which lowers the bound of $L_{0\text{-}1}(f_n)$.

More intuitively, ChunkKV focus on reducing the noise on some complete training examples, but some other examples overall with low importance will be discarded. Then, the model identifies the $x_{\text{test}}$ from those clean and more related training examples $O_i$ and neglect those $O_i$ with less importance.

Note that here, we do not provide the rigorous proof on how KV cache sparsity enhances the requirement of the distinguishability and how different $\text{KL}_j(\theta^\star||\theta)$ on $O_i = [x_i, y_i]$ influences the bound $L_{0\text{-}1}(f_n)$. We left this as the future work to analyze how KV cache sparsity influences the in-context learning.

# D  Additional Related Work

**Chunking Method.** The chunking methodology is widely used in the field of NLP due to its simplicity and effectiveness [52]. In the era of LLMs, chunking is primarily applied in data pre-processing. For example, Shi et al. [53] suggest grouping related training data into chunks to achieve better convergence curves to pre-train LLMs. Fei et al. [54] apply a topic-based chunking method to improve the semantic compression of prompts. Furthermore, chunking plays an important role in the Retrieval-Augmented Generation (RAG) field [55–57]. It serves to divide documents into units of information with semantic content suitable for embedding-based retrieval and processing by LLMs.

**Layer-Wise Technique.** The layer-wise technique is widely used in the training and inference of large language models (LLMs). LISA [58] is a layer-wise sampling method based on observations of the training dynamics of Low-Rank Adaptation (LoRA)[35, 36, 38, 59, 60] across layers. LAMB[61] is a layer-wise adaptive learning rate method that speeds up LLM training by stabilizing training convergence with large batch sizes. DoLa [62] employs layer-wise contrasting to reduce hallucinations during LLM inference.

**KV Cache Sharing.** Recent work has explored various strategies for sharing KV caches across transformer layers. Layer-Condensed KV Cache (LCKV) [63] computes KVs only for the top layer and pairs them with queries from all layers, while optionally retaining standard attention for a few top and bottom layers to mitigate performance degradation. Similarly, You Only Cache Once (YOCO) [64] computes KVs exclusively for the top layer but pairs them with queries from only the top half of layers, employing efficient attention in the bottom layers to maintain a constant cache size. In contrast, Cross-Layer Attention (CLA) [65] divides layers into groups, pairing queries from all layers in each group with KVs from that group's bottom layer. MiniCache [66] introduces a novel method that merges layer-wise KV caches while enabling recovery during compute-in-place operations, optimizing KV cache size. These methods illustrate various trade-offs between computation, memory usage, and model performance when sharing KV caches across transformer layers.

---

$^\dagger$Here, training means prompt learning [24].

**Long-Context Benchmarks.** The landscape of long-context model benchmarks has evolved to encompass a wide range of tasks, with particular emphasis on retrieval and comprehension capabilities. Benchmarks for understanding have made significant strides, with $\infty$-Bench [67] pushing the boundaries by presenting challenges that involve more than 100,000 tokens. LongBench [25] has introduced bilingual evaluations, addressing tasks such as long-document question answering, summarization, and code completion. Complementing these efforts, ZeroSCROLLS [68] and L-Eval [69] have broadened the scope to include a diverse array of practical natural language tasks, including query-driven summarization.

In parallel, retrieval benchmarks have largely relied on synthetic datasets, offering researchers precise control over variables such as the length of input tokens. This approach minimizes the impact of disparate parametric knowledge resulting from varied training methodologies. A significant body of recent work has concentrated on the development of synthetic tasks specifically for retrieval evaluation [26, 70–73]. In addition, researchers have explored the potential of extended contexts in facilitating various forms of reasoning [74]. Liu et al. [75] explores the influence of the KV cache method beyond long-context benchmarks. LongGenBench [76] advances beyond traditional benchmarks by redesigning the format of questions and necessitating that LLMs respond with a single, cohesive long-context answer.

This dual focus on synthetic retrieval tasks and comprehensive understanding benchmarks reflects the field's commitment to rigorously assessing the capabilities of long-context models across diverse linguistic challenges.

**Prompting Compression.** In the field of prompt compression, various designs effectively combine semantic information to compress natural language. [77] utilize soft prompts to encode more information with fewer tokens. [78] present AutoCompressor, which uses soft prompts to compress the input sequence and extend the original length of the base model. Both [79] and [80] recurrently apply LLMs to summarize input texts, maintaining long short-term memory for specific purposes such as story writing and dialogue generation. The LLMLingua series [54, 81, 82] explores the potential of compressing LLM prompts in long-context, reasoning, and RAG scenarios. [54] use pre-trained language models to chunk the long context and summarize semantic information, compressing the original context.

**KV Cache Quantization.** Among quantization approaches for KV cache optimization, SmoothQuant [83] stands out as a notable post-training technique. By implementing balanced transformations between activation and weight quantization difficulties, this method enables 8-bit KV cache compression with minimal performance degradation. However, research by [84] has demonstrated that SmoothQuant encounters substantial accuracy deterioration when compressed beyond 8 bits. In parallel developments, the FlexGen system [85] implements group-wise quantization at 4 bits for both key and value caches, while KIVI [45] introduces an innovative approach with asymmetric 2-bit per-channel quantization that requires no additional tuning phase. Additionally, AnTKV [86] introduces a two-stage token-aware vector quantization framework, leveraging gradient-weighted centroid learning and anchor-token selection to achieve ultra-low-bit KV cache quantization with minimal accuracy loss.

# E   Statistics of Models

Table 27 provides configuration parameters for LLMs that we evaluated in our experiments.

| Model Name | LLaMA-3-8B-Instruct | Mistral-7B-Instruct-v0.2 & 0.3 | Qwen2-7B-Instruct |
|---|---|---|---|
| $L$ (Number of layers) | 32 | 32 | 28 |
| $N$ (Number of attention heads) | 32 | 32 | 28 |
| $D$ (Dimension of each head) | 128 | 128 | 128 |

Table 27: Models Configuration Parameters

# F  Statistics of Datasets

LongBench is a meticulously designed benchmark suite that evaluates the capabilities of language models in handling extended documents and complex information sequences. This benchmark was created for multi-task evaluation of long-context inputs and includes 17 datasets covering tasks such as single-document QA [87, 88], multi-document QA [89–92], summarization [93–96], few-shot learning [97–99], synthetic tasks and code generation [100, 101]. The datasets feature an average input length ranging from $1K$ to $18K$ tokens, requiring substantial memory for KV cache management.

Table 28 shows the statistics of the datasets that we used in our experiments.

| DATASET | # TRAIN | # TEST |
|---|---|---|
| GSM8K [27] | 7,473 | 1,319 |
| LONGBENCH [25] | - | 4,750 |
| NIAH* [26] | - | 800 |
| JAILBREAKV [28] | - | 28,000 |

Table 28: Dataset Statistics. # TRAIN and # TEST represent the number of training and test samples, respectively. *: The size of the NIAH test set varies based on the context length and step size, typically around 800 samples per evaluation.

# G  Prompt

Table 29 shows the prompt for the Figure 1.

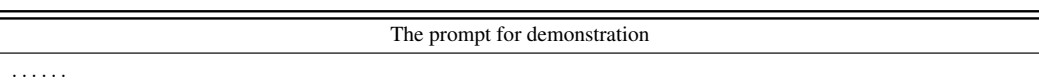

| The prompt for demonstration |
|---|
| . . . . . .
. . . . . .
The purple-crested turaco (Gallirex porphyreolophus) or, in South Africa, the purple-crested loerie, (Khurukhuru in the Luvenḍa (Venḍa) language) is a species of bird in the clade Turaco with an unresolved phylogenetic placement. Initial analyses placed the purple-crested turaco in the family Musophagidae, but studies have indicated that these birds do not belong to this family and have been placed in the clade of Turacos with an unresolved phylogeny. It is the National Bird of the Kingdom of Eswatini, and the crimson flight feathers of this and related turaco species are important in the ceremonial regalia of the Swazi royal family. This bird has a purple-coloured crest above a green head, a red ring around their eyes, and a black bill. The neck and chest are green and brown. The rest of the body is purple, with red flight feathers. Purple-crested turacos are often seen near water sources, where they can be observed drinking and bathing, which helps them maintain their vibrant plumage. Purple-crested turacos are considered to be large frugivores that are known to carry cycad seeds from various plant species long distances from feeding to nesting sites. After fruit consumption, they regurgitate the seeds intact where they can germinate nearby. G. porphyreolophus primarily consumes fruits whole like many other large frugivores which are suggested to be necessary for effective ecosystem functioning. Among similar turacos, the purple-crested turaco have faster minimum transit times when consuming smaller seed diets than larger seed diets, and G. porphyreolophus has been shown to have significantly faster pulp (seedless fruit masses) transit time than another closely related Turaco when fed only the pulp of larger-seeding fruits than smaller-seeding fruits. In addition to their frugivorous diet, these birds are occasionally seen foraging for other food items such as nuts and leaves, which provide essential nutrients. They are also known to coexist with various other animals, including those that might enjoy strawberries and other similar fruits. The purple-crested turaco's role in seed dispersal is crucial, and their interaction with different elements of their habitat, including water and diverse plant materials, highlights their importance in maintaining ecological balance.
. . . . . .
. . . . . . |

Table 29: The prompt for demonstration

Here we provide the CoT prompt exemplars for GSM8K which is used in section 4.1.

| GSM8K experiemnt CoT Prompt Exemplars |
| --- |
| Question: There are 15 trees in the grove. Grove workers will plant trees in the grove today. After they are done, there will be 21 trees. How many trees did the grove workers plant today? |
| There are 15 trees originally. Then there were 21 trees after some more were planted. So there must have been 21 - 15 = 6. The answer is 6. |
| Question: If there are 3 cars in the parking lot and 2 more cars arrive, how many cars are in the parking lot? |
| There are originally 3 cars. 2 more cars arrive. 3 + 2 = 5. The answer is 5. |
| Question: Leah had 32 chocolates and her sister had 42. If they ate 35, how many pieces do they have left in total? |
| Originally, Leah had 32 chocolates. Her sister had 42. So in total they had 32 + 42 = 74. After eating 35, they had 74 - 35 = 39. The answer is 39. |
| Question: Jason had 20 lollipops. He gave Denny some lollipops. Now Jason has 12 lollipops. How many lollipops did Jason give to Denny? |
| Jason started with 20 lollipops. Then he had 12 after giving some to Denny. So he gave Denny 20 - 12 = 8. The answer is 8. |
| Question: Shawn has five toys. For Christmas, he got two toys each from his mom and dad. How many toys does he have now? |
| Shawn started with 5 toys. If he got 2 toys each from his mom and dad, then that is 4 more toys. 5 + 4 = 9. The answer is 9. |
| Question: There were nine computers in the server room. Five more computers were installed each day, from monday to thursday. How many computers are now in the server room? |
| There were originally 9 computers. For each of 4 days, 5 more computers were added. So 5 * 4 = 20 computers were added. 9 + 20 is 29. The answer is 29. |
| Question: Michael had 58 golf balls. On tuesday, he lost 23 golf balls. On wednesday, he lost 2 more. How many golf balls did he have at the end of wednesday? |
| Michael started with 58 golf balls. After losing 23 on tuesday, he had 58 - 23 = 35. After losing 2 more, he had 35 - 2 = 33 golf balls. The answer is 33. |
| Question: Olivia has $23. She bought five bagels for $3 each. How much money does she have left? |
| Olivia had 23 dollars. 5 bagels for 3 dollars each will be 5 x 3 = 15 dollars. So she has 23 - 15 dollars left. 23 - 15 is 8. The answer is 8. |

Table 30: GSM8K CoT Prompt Exemplars

## H    Impact Statement

Our study does not involve human subjects, data collection from individuals, or experiments on protected groups. The models and datasets used in this work are publicly available and widely used in the research community. We have made efforts to ensure our experimental design and reporting of results are fair, unbiased, and do not misrepresent the capabilities or limitations of the methods presented.

In our work on KV cache compression for large language models, we acknowledge the potential broader impacts of improving efficiency in AI systems. While our method aims to reduce computational resources and potentially increase accessibility of these models, we recognize that more efficient language models could also lead to increased deployment and usage, which may have both positive and negative societal implications. We encourage further research and discussion on the responsible development and application of such technologies.

We declare no conflicts of interest that could inappropriately influence our work. All experiments were conducted using publicly available resources, and our code will be made available to ensure reproducibility. We have made every effort to cite relevant prior work appropriately and to accurately represent our contributions in the context of existing research.

## I    Limitations

While ChunkKV demonstrates strong performance and efficiency, we acknowledge several limitations that provide avenues for future research.

First, the core design of ChunkKV prioritizes the preservation of semantic coherence by treating chunks as indivisible units. This approach may be less suitable for tasks requiring absolute semantic fidelity, where every single token can be critical. For example, in domains such as legal or biomedical document analysis, discarding any part of the text, even if it appears less relevant based on attention scores, could lead to the loss of crucial information. Our method is therefore optimized for tasks where capturing the core gist is more important than verbatim retention of every detail.

Second, our experiments with a hybrid compression strategy (using ChunkKV in early layers and a token-level method in deeper layers) revealed a nuanced, task-dependent performance trade-off. While pure ChunkKV excelled in local information retrieval tasks (e.g., Single- and Multi-Document QA), the hybrid model showed surprising strength in tasks requiring global understanding (e.g., Summarization). This suggests that a one-size-fits-all compression strategy may not be optimal for all scenarios. The ideal approach might involve dynamically adapting the compression granularity based on the perceived task requirements.

Finally, our current implementation relies on fixed-size chunks for computational efficiency. While our ablations show this is a robust strategy, adaptively determining chunk boundaries based on linguistic cues (e.g., sentence endings) could potentially improve semantic integrity further, although this would introduce additional inference latency that must be carefully managed.

# J  Licenses

For the evaluation dataset, all the datasets, including, GSM8K [27], LongBench [25] are released under MIT license. NIAH [26] is released under GPL-3.0 license.

