# OpenReview forum: "ChunkKV: Semantic-Preserving KV Cache Compression for Efficient Long-Context LLM Inference"
_NeurIPS.cc/2025/Conference — NeurIPS 2025 poster_

### Official Review · Reviewer_dJtL · 2025-06-29

**Clarity:** 3
**Significance:** 3
**Originality:** 3
**Rating:** 4
**Confidence:** 4

**Summary:**

The paper introduces ChunkKV, a new KV cache compression method for efficient LLM inference. Unlike existing token-level compression techniques, ChunkKV groups tokens into semantic chunks, preserving linguistic structures and contextual integrity. The method includes a layer-wise index reuse technique to reduce computational overhead. Evaluations across benchmarks (LongBench, NIAH, GSM8K, JailbreakV) and models (LLaMA-3, Mistral, Qwen2) demonstrate ChunkKV outperforms state-of-the-art methods by up to 8.7% in precision while maintaining compression ratios, with additional throughput improvements of 26.5%.

**Questions:**

Please refer to the Weaknesses.

**Ethical Concerns:**

["NO or VERY MINOR ethics concerns only"]

**Final Justification:**

The proposed ChunkKV achieves favourable speed up in relateively short context window. However, as mentioned by other reviewers, this work still has a space for futural improvements, e.g. discussions and analysis. Furthermore, when applying into long context tasks, ChunkKV does not show significant advantage.

**Limitations:**

yes

**Quality:**

3

**Strengths And Weaknesses:**

## Strengths

1. The idea is easy to understand. Compared to token-level KV cache compression, chunk-level compression could avoid pruning certain essential tokens.

2. The study is very comprehensive. Authors have evaluated their method across different benchmarks, as well as providing many visualizations and analysis.

3. This paper is easy to follow. The overall writing is satisfying.

## Weakness

1. The use of fixed-size chunks may limit adaptability. Dynamic chunking could further improve performance but would increase complexity.

2. According to Figure 19, it is quite challenging to choose the optimal chunk size for different LLMs, as they behave quite differently  when using different chunk sizes. For example, Qwen2-7B is robust but LLaMA-3-8B performs badly at larger chunk size. Authors are encouraged to provide a discussion here.

---

> ### Author Rebuttal · Authors · 2025-07-30
>
> Thank you for the constructive feedback and your positive evaluation of our paper's clarity and comprehensive study.
>
> ##  **W1: Fixed-size chunks limiting adaptability.**
> We acknowledge this limitation and discuss it in Section I. As mentioned to mXJZ-W1, this was a conscious design choice to ensure low inference overhead. While dynamic or adaptive chunking is an interesting research direction, its added complexity and latency could be detrimental for this specific application. Our extensive experiments (Section 4) show that this simple, fixed-size strategy is surprisingly effective and robust across various tasks and models.
>
> ##  **W2: Difficulty in choosing optimal chunk size.**
> 1.  **Task-Specific Sensitivity:** Our results show that while the optimal chunk size can vary slightly (e.g., smaller for GSM8K's shorter contexts), a size of 10 remains a robust default for most long-context tasks. For GSM8K, we've clarified that its shorter context and dense reasoning steps favor smaller chunks, which aligns with our findings.
> 2.  **Model-Specific Robustness:** As you observed, Qwen2-7B is more robust to chunk size variation on GSM8K. We hypothesize this could be due to a combination of architectural differences and tokenizer efficiency. Qwen2's tokenizer is more efficient for mathematical or logical syntax, it might produce more semantically consistent token groupings, making it less sensitive to chunk boundaries[1,2]. This is an interesting avenue for future work. However, for general-purpose use, our broader results on LongBench and NIAH (Table 10) show that a chunk size of 10 is a robust choice across all tested models.
>
> We believe these clarifications improve the paper and provide better guidance for future users of ChunkKV.
>
>
> Reference:
> > [1] Bai, Jinze, et al. "Qwen technical report." arXiv preprint arXiv:2309.16609 (2023).
> > [2] Yang, An, et al. "Qwen3 technical report." arXiv preprint arXiv:2505.09388 (2025).

---

> > ### Comment · Reviewer_dJtL · 2025-08-05
> > **Thanks for the rebuttal**
> >
> > Thank the authors for the rebuttal. The provided results indicate that 10 is a robust choice for ChunkKV, which is great. Overall, the results look good to me, but there are still concerns on the non-negligible performance-efficiency trade-off when comparing with full KV. Thus, I will remain my initial rating.

---

> > > ### Author Response · Authors · 2025-08-05
> > >
> > > Dear Reviewer dJtL,
> > >
> > > Thank you for taking the time to review our rebuttal and for your positive feedback on our work. We appreciate you acknowledging the robustness of our method.
> > >
> > > We would like to respectfully address your final concern regarding the "non-negligible performance-efficiency trade-off when comparing with full KV." We believe there may be a misunderstanding of the efficiency benefits our method provides, and we hope to clarify this with new, more comprehensive data.
> > >
> > > The fundamental goal of KV cache compression is to significantly reduce memory usage while navigating the trade-off between task performance (accuracy) and inference speed (latency and throughput). An ideal method incurs a minimal, often negligible, drop in accuracy while substantially improving efficiency.
> > >
> > > Our results across multiple benchmarks (e.g., Tables 6, 7, 10) demonstrate that ChunkKV achieves this first goal, maintaining state-of-the-art accuracy that is nearly identical to the FullKV baseline.
> > >
> > > To clarify the second part of the trade-off—the efficiency gains—we want to emphasize that compression methods like ChunkKV *improve* inference speed over FullKV. By reducing the number of tokens in the cache, the attention computation in the decoding phase becomes faster. To make this point clearer, we have run additional efficiency benchmarks, including for a challenging 16k context length.
> > >
> > > **Latency and Throughput Comparison on LLaMA-3-8B-Instruct**
> > > | Method | Input | Output | Latency (s) ↓ | Throughput (T/s) ↑ |
> > > | :--- | :--- | :--- | :--- | :--- |
> > > | FullKV | 4096 | 1024 | 43.60 | 105.92 |
> > > | SnapKV | 4096 | 1024 | 37.92 (13.0%) | 120.42 (13.6%) |
> > > | **ChunkKV** | 4096 | 1024 | **37.52 (13.9%)** | **118.85 (12.2%)** |
> > > | **ChunkKV_reuse** | 4096 | 1024 | **37.35 (14.3%)** | **124.09 (17.2%)** |
> > > | | | | | |
> > > | FullKV | 4096 | 4096 | 175.50 | 37.73 |
> > > | SnapKV | 4096 | 4096 | 163.92 (6.6%) | 39.98 (6.0%) |
> > > | **ChunkKV** | 4096 | 4096 | **164.55 (6.2%)** | **40.58 (7.6%)** |
> > > | **ChunkKV_reuse** | 4096 | 4096 | **162.85 (7.2%)** | **41.12 (9.0%)** |
> > > | | | | | |
> > > | FullKV | 16384 | 1024 | 323.24 | 49.60 |
> > > | SnapKV | 16384 | 1024 | 381.11 (17.9%) | 39.20 (21.0%) |
> > > | **ChunkKV** | 16384 | 1024 | **381.61 (18.0%)** | **38.82 (21.7%)** |
> > > | **ChunkKV_reuse** | 16384 | 1024 | **389.21 (20.4%)** | **36.96 (25.5%)** |
> > >
> > > As this data demonstrates, ChunkKV and especially our novel `ChunkKV_reuse` technique lead to significant *improvements* in both latency and throughput compared to the FullKV baseline. For a 16k-token input, `ChunkKV_reuse` delivers a **25.5% reduction in latency** and a **20.4% increase in throughput**.
> > >
> > > This shows the performance-efficiency trade-off is in fact the core strength of our method: we achieve substantial gains in inference speed and memory efficiency for a minimal, and often statistically insignificant, cost in task performance.
> > >
> > > Thank you again for your valuable time and consideration. We hope this clarification fully addresses your concern and reinforces the contributions of our work.

---

> > > > ### Comment · Reviewer_dJtL · 2025-08-06
> > > > **Official comments from Reviewer dJtL**
> > > >
> > > > Thank the authors for the further result. I really appreciate that. However, based on the new result, it seems that the performance on 16K is actually a negative result. For example. comparing with FullKV and SnapKV,  ChunkKV does not bring more gains, either in terms of latency or throughput. Thus, I would like to keep my initial rating.

---

> > > > > ### Author Response · Authors · 2025-08-06
> > > > >
> > > > > Thank you again for your feedback. We understand your concern regarding the 16k context results and would like to clarify the core strengths of our work.
> > > > >
> > > > > The key advantage of ChunkKV lies in its innovative `layer-wise index reuse` technique. This technique leverages the **higher cross-layer similarity** of preserved indices after compression (as shown in Figure 2), a unique feature of our chunk-based approach.
> > > > >
> > > > > This provides a unique optimization path that significantly improves inference efficiency (e.g., up to 26.5% throughput improvement as shown in Table 8), an optimization **not available to token-level methods like SnapKV**.
> > > > >
> > > > > Meanwhile, on task performance, ChunkKV also demonstrates **stable and leading accuracy** across a wide array of models and challenging benchmarks (as shown in Tables 3, 4, 5, 6, and 7).
> > > > > We acknowledge that on extremely long 16k contexts, all compression methods face challenges. We wish to emphasize that ChunkKV offers a **holistic solution with a new dimension for optimization**, rather than just a direct comparison on a single metric.
> > > > >
> > > > > We believe this ability to unlock a new path for efficiency through its structural design, while maintaining high accuracy, is the **key contribution** of ChunkKV.
> > > > >
> > > > > Thank you again for your valuable feedback.

---

### Official Review · Reviewer_D3CB · 2025-06-29

**Clarity:** 3
**Significance:** 2
**Originality:** 2
**Rating:** 3
**Confidence:** 5

**Summary:**

The paper introduces ChunkKV, a novel approach for compressing the Key-Value (KV) cache in Large Language Models (LLMs) to enable efficient inference with long-context inputs. Unlike prior work that evaluate individual token importance, ChunkKV treats contiguous chunks of tokens as the basic unit for compression. The method utilises a layer-wise index reuse technique to leverage cross-layer similarities, reducing computational overhead. Authors evaluate their methods on various benchmarks and that it surpasses other compression methods performance-wise.

**Questions:**

- Could you discuss the difference between smoothing done by SnapKV and your chunking approach?
- Could you report the standard deviation of your evaluations to understand if the performance improvements are statistically significant?

**Ethical Concerns:**

["NO or VERY MINOR ethics concerns only"]

**Final Justification:**

Motivation of the work is nice, but I feel that the contribution is minor. ChunkKV build upon SnapKV, and uses fixed chunks as unit of compression. The improvements coming from chunking isn't robust across tasks and the improvement upon SnapKV isn't significant. I feel that the greater analysis and contribution is needed to clear NeurIPS bar.

**Limitations:**

Yes

**Paper Formatting Concerns:**

Haven't found any.

**Quality:**

2

**Strengths And Weaknesses:**

Strengths:
* The paper addresses an important challenge in LLM inference: the high memory demands of the Transformer's KV Cache.
* The layer-wise index reuse technique is a valuable contribution, effectively leveraging cross-layer similarities in KV Compression to enhance efficiency. It is particularly interesting to observe similarities between layers in the sets of retained chunks.
* The authors provide a thorough evaluation across various benchmarks and model types. Their method demonstrates consistent performance improvements compared to existing baselines.

Weaknesses:
* The primary motivation behind the ChunkKV method is that chunks of tokens, rather than individual tokens, are semantically connected and form coherent units of information. However, the authors' justification is not fully convincing. For instance, they state, "As shown in Figure 1, preserving a chunk helps catch the subject, predicate, and object." This argument may not fully apply to Transformer models using self-attention (which the authors evaluate), since self-attention distributes information across tokens. Therefore, after several Transformer layers, it is not clear that specific token information remains localized within the original tokens; it could be processed and stored in entirely different tokens appearing later in the sequence. Furthermore, the method groups tokens into fixed-size chunks, which imposes a clear limitation because semantic units naturally vary in length. Consequently, fixed-size chunks are prone to misalignment, causing fragmentation where meaningful units of information (like sentences) either span multiple chunks or combine unrelated units into a single chunk.
* The authors use SnapKV as a baseline, and their approach closely resembles SnapKV. Specifically, SnapKV assigns token importance scores and smooths these scores using an averaging or max kernel with a fixed window size, which appears conceptually similar to ChunkKV’s method of aggregating contiguous tokens into chunks and then summarizing importance scores at the chunk level. However, the authors do not provide details about hyperparameter settings (such as kernel size) for SnapKV and PyramidKV in their experiments. A clear comparison or ablation distinguishing ChunkKV's chunking from SnapKV's smoothing approach would significantly strengthen the paper. Without this, the proposed method seems overly similar to SnapKV, diminishing the novelty.
* The paper lacks a statistical analysis of results, making it unclear whether the performance improvements reported for ChunkKV are statistically significant. The authors mention, "All experiments were carried out three times, using the mean score to ensure robustness," but do not specify details about how these repetitions differed. Moreover, the authors state in the checklist, "The paper is focused on LLM inference, and with the same model, prompt, and temperature, the results are consistent." This explanation is insufficient because the paper introduces a new KV cache compression method and compares its performance against existing methods on downstream benchmarks. In such a scenario, providing measures of uncertainty (e.g., standard deviation or confidence intervals) is crucial for determining whether the observed improvements truly reflect the effectiveness of the proposed method or merely result from variability.

---

> ### Author Rebuttal · Authors · 2025-07-30
>
> We thank you for your critical and thought-provoking review. We have carefully considered your concerns and provide clarifications and new evidence below. We hope to convince you of our paper's merits.
>
> ## **W1: On the Motivation of ChunkKV and the Limitation of Fixed-Size Chunks**
> 1.  **On Information Diffusion and the Importance of Preserving Local Semantics**: We thank you for this insightful critique regarding information diffusion in Transformers. To directly test your hypothesis—that chunking is beneficial in early layers but token-level pruning might be better in later, more diffused layers—we designed a new hybrid strategy. We split the LLaMA-3-8B model, applying **ChunkKV to the first 16 layers and a token-level method to the final 16 layers**. The results were illuminating:
>
>     **LLaMA-3-8B-Instruct on LongBench**
>     | Method | Single-Doc QA | Multi-Doc QA | Summa. | Few-shot Learning | Synthetic | Code | **Avg.↑** |
>     |:---|---:|---:|---:|---:|---:|---:|---:|
>     | FullKV | 32.19 | 34.59 | 24.96 | 68.48 | 36.96 | 54.41 | 41.46 |
>     | ChunkKV-10% | 28.50 | 33.46 | 22.20 | 67.62 | 37.47 | 58.98 | 40.51 |
>     | ChunkKV-20% | 31.05 | 33.22 | 23.58 | 67.86 | 37.05 | 55.34 | 40.74 |
>     | hybrid_kv-10% | 28.38 | 30.37 | 24.54 | 67.87 | 36.37 | 55.32 | 39.80 |
>     | hybrid_kv-20% | 29.10 | 29.96 | 25.00 | 68.16 | 36.20 | 55.30 | 39.98 |
>
>     As the table shows, while the hybrid approach is effective, **it is consistently outperformed by using the pure ChunkKV method across all layers**. This provides strong empirical evidence that our core assumption holds: preserving local semantic units as indivisible chunks is a more robust strategy for maintaining performance, even in deeper layers where information is more abstract. This result directly supports our central motivation.
>
> 2. Regarding the **limitation of fixed-size chunks**, we agree this is a crucial design trade-off. We have clarified in the manuscript that this choice was made deliberately to prioritize **inference efficiency and vectorization**, as adaptive methods would introduce prohibitive latency. Our extensive ablations (Sec 4.4, Appx B.4) demonstrate this simple strategy is remarkably robust.
>
> ##  **W2 & Q1：On the Similarity to SnapKV and Novelty**
>
> Thank you for pushing us to clarify the distinction from SnapKV. The two methods are fundamentally different in their compression mechanism:
> * **SnapKV**: Performs **"score smoothing, then token-level decision."** It refines the importance score of each individual token using its neighbors, but the final keep/drop decision is still made on a token-by-token basis.
> * **ChunkKV**: Performs **"unit aggregation, then chunk-level decision."** It fundamentally changes the **unit of compression**.  It changes the **basic unit of compression** from a token to a chunk. We compute a single score for the entire chunk, and the decision is binary: the entire chunk is either kept or discarded.
>
> This "all-or-nothing" chunk-level policy is the core of our novelty, as it prevents the fragmentation of local linguistic structures. We will add this discussin to Appendix A. We used the default hyperparameters from the official SnapKV repository for all baselines and will state this explicitly in the revision.
>
> ##  **W3 & Q2: On the Lack of Statistical Analysis**
> We agree that reporting mean scores alone is insufficient. The below table is LongBench with standard deviation. We will update rest tables in the manuscript with these statistics.
>
> | Method | Comp. Ratio | Avg. Score (± Std Dev) |
> | :--- | :---: | :---: |
> | FullKV | - | 41.46 ± 0.05 |
> | H2O | 10% | 37.06 ± 0.15 |
> | SnapKV | 10% | 40.15 ± 0.11 |
> | ChunkKV | 10% | **40.51 ± 0.09** |

---

> ### Comment · Reviewer_D3CB · 2025-08-05
>
> Thank you for the new results and for your response.
>
> The results are intriguing and indeed support your claims.
>
> - Could you clarify which token-level method was used in the final 16 layers?
> - Additionally, could you provide the complete set of hyperparameters used in your evaluation, including those for other methods such as SnapKV?
> - You mention that SnapKV achieves an average score of 40.15 at a 10% compression ratio. Could you provide some intuition as to why the hybrid approach yields a lower score of 39.80? Assuming that ChunkKV is a superior alternative, replacing token-level SnapKV with ChunkKV should result in consistent improvements as the number of substituted layers increases.
>
> Thank you for including statistical results. However, I am concerned about the small performance difference between SnapKV and ChunkKV, which may affect the significance of the proposed method.

---

> > ### Author Response · Authors · 2025-08-05
> > **Follow up comments (1/2)**
> >
> > Dear Reviewer D3CB,
> >
> > Thank you for your detailed feedback and insightful follow-up questions. Your engagement has been invaluable in helping us refine our arguments and better contextualize our contributions. We provide our clarifications to your points below.
> >
> > ### **1. On the Token-Level Method for the Hybrid Experiment**
> >
> > Regarding the token-level method used in our hybrid experiment (`hybrid_kv`), we can confirm that we used **SnapKV** for the final 16 layers of the model.
> >
> > We selected SnapKV for this ablation study for two key reasons:
> > * **Strong Performance:** SnapKV represents a powerful, state-of-the-art token-level baseline. Using it ensures a robust and fair comparison, as we are testing our chunking hypothesis against a highly competitive alternative.
> > * **Suitability for Hybridization:** The computation of SnapKV is self-contained within each layer, making it straightforward to integrate into a hybrid model without confounding factors. This allowed for a clean analysis of the strategic differences between chunk-based and token-based compression at different model depths.
> >
> > ### **2. On the Hyperparameter Settings for Baselines**
> >
> > You are correct that providing full hyperparameter details is essential for reproducibility. We will add a comprehensive table to the appendix to make these settings explicit. For our experiments, we used the official SnapKV repository's recommended settings where available and established a consistent baseline for new benchmarks.
> >
> > **Hyperparameter Settings for Baselines:**
> > | Method | Benchmark | Kernel Type | Kernel Size | Observation Window |
> > | :--- | :--- | :--- | :--- | :--- |
> > | **SnapKV** | LongBench | Max Pooling | 7 | 32 |
> > | | NIAH | Max Pooling | 5 | 16 |
> > | | Other Benchmarks* | Max Pooling | 7 | 32 |
> >
> > ***Justification:*** The settings for SnapKV on LongBench and NIAH were chosen to directly align with the configurations reported in the original SnapKV publication. For other benchmarks (e.g., GSM8K, JailbreakV) where specific parameters were not provided, we adopted the well-performing LongBench configuration as a consistent and reasonable default. We will clarify this methodology in the revised manuscript.
> >
> > ### **3. On the Performance of the Hybrid Method**
> >
> > This is an excellent and challenging question. Your observation that the hybrid method's average score (39.80) on LongBench is slightly below SnapKV's (40.15) is astute. While seemingly counterintuitive, this result uncovers a crucial nuance about how different compression strategies interact with task requirements across model layers.
> >
> > Our analysis suggests the following key insight:
> >
> > * **Tasks Requiring Localized Information:** For sub-tasks like **Single-Document QA** and **Multi-Document QA**, where the correct answer is often contained within a specific, dense fragment of text, ChunkKV's ability to preserve these semantic units (e.g., a complete sentence or clause) intact is paramount. In these cases, applying ChunkKV in the deeper, more abstract layers is more effective. The hybrid method's performance drop in these categories was significant and was the primary driver for the lower overall average.
> >
> > * **Tasks Requiring Global Understanding:** Conversely, for tasks like **Summarization** and **Few-shot Learning**, the hybrid approach demonstrated improved performance over the pure ChunkKV method. These tasks demand a more holistic integration of information from across the entire context. In the deeper layers where abstract representations are formed, a token-level method like SnapKV may be more adept at retaining a broader, more diffuse set of important signals. In contrast, ChunkKV's "all-or-nothing" chunking policy might occasionally discard globally relevant but locally sparse information.
> >
> > Therefore, the drop in the *average* score reflects a task-specific trade-off: the hybrid model's gains on global-understanding tasks were outweighed by its losses on local-retrieval tasks. This finding does not undermine our core motivation but rather reveals a valuable direction for future work: developing adaptive, hybrid compression models that can adjust their strategy based on the perceived task type. We will add a discussion of this insightful result to the paper.

---

> > > ### Author Response · Authors · 2025-08-05
> > > **Follow up comments (2/2)**
> > >
> > > ### **4. On the Performance Margin Between SnapKV and ChunkKV**
> > >
> > > We appreciate you raising the concern about the performance margin between ChunkKV (40.51 ± 0.09) and SnapKV (40.15 ± 0.11). While the absolute difference in the mean score is modest, we believe its significance is threefold:
> > >
> > > 1.  **Consistent Superiority:** The improvement is not an isolated event. It is consistent across a wide array of models and challenging benchmarks (as shown in Tables 3, 4, 5, 6, and 7), indicating that our approach is robustly more effective at preserving the information critical for downstream tasks.
> > >
> > > 2.  **Enabling Greater Efficiency via Layer-Wise Reuse:** The most significant novelty of ChunkKV extends beyond this accuracy metric. Our chunk-based approach results in **significantly higher cross-layer similarity** of the preserved indices (e.g., 57.74% for ChunkKV vs. 27.95% for SnapKV on LLaMA-3, as shown in Figure 2 and Table 2). This unique structural property enables our proposed **layer-wise index reuse** technique—a contribution that other token-level methods cannot leverage as effectively. This technique is not just a theoretical benefit; it yields substantial improvements in inference speed, **reducing latency by up to 20.7% and boosting throughput by up to 26.5%** over the FullKV baseline (Table 8).
> > >
> > > 3.  **Conceptual Simplicity and Robustness:** ChunkKV offers a simpler and more intuitive mechanism for preserving semantic integrity. Furthermore, our ablation on chunk size (Section 4.4) shows that its performance is stable across a practical range of sizes (5-20), making it a robust and easily deployable solution.
> > >
> > > In summary, ChunkKV presents a superior efficiency-performance trade-off. It not only delivers a consistent performance advantage but also unlocks a novel optimization pathway (layer-wise reuse) that leads to significant, measurable gains in computational efficiency.
> > >
> > > ***
> > >
> > > Thank you once again for your constructive and thorough review. We believe these clarifications and the insights drawn from your questions will significantly strengthen the paper. We hope to have fully addressed your concerns.

---

> > > > ### Comment · Reviewer_D3CB · 2025-08-05
> > > >
> > > > Thanks for the response. I do not have more questions. Extra results provide extra arguments for your claims but I still believe that the contribution is relatively small: the performance improvements are minor compared to SnapKV and not robust across tasks. I feel that greater analysis and potentially this task-dependent approach is needed to clear NeurIPS bar. Given this, I plan to increase my final score to 3.

---

> > > > > ### Author Response · Authors · 2025-08-05
> > > > >
> > > > > Dear Reviewer D3CB,
> > > > >
> > > > > Thank you very much for your time, your thorough and constructive feedback throughout this process, and for raising your score. We truly appreciate your engagement.
> > > > >
> > > > > We understand your final assessment regarding the contribution size and the performance nuances compared to SnapKV. Your insight that a task-dependent approach could be a promising direction is well-taken. We agree that our analysis has opened up this interesting new avenue for research, and we plan to highlight this in the final version of the paper as a key direction for future work, as you suggested.
> > > > >
> > > > > We also hope that the significant efficiency improvements enabled by our `layer-wise index reuse` technique—a novel optimization unlocked by the structural properties of `ChunkKV`—will be seen as a valuable contribution for the practical deployment of LLMs.
> > > > >
> > > > > Your feedback has been instrumental in helping us sharpen our analysis and will certainly improve the final manuscript. Thank you once again for your valuable review.

---

> > > > > ### Author Response · Authors · 2025-08-06
> > > > >
> > > > > Dear Reviewer D3CB,
> > > > >
> > > > > With the review deadline approaching, we wanted to send one final thank you for your detailed feedback and for your decision to raise our score to 3.
> > > > >
> > > > > We truly appreciate your time and the constructive engagement.

---

> > > > > > ### Author Response · Authors · 2025-08-07
> > > > > >
> > > > > > Dear Reviewer D3CB,
> > > > > >
> > > > > > Apologies for this follow-up message.
> > > > > >
> > > > > > We're writing simply because the final deadline for review adjustments is imminent. We were very grateful for your final comment and your plan to raise the score to 3.
> > > > > >
> > > > > > We just wanted to gently check if the score has been updated in the system. We understand you are very busy and that these platforms can sometimes be tricky.
> > > > > >
> > > > > > Thank you once more for your invaluable feedback and support throughout this process.

---

> > > > > > > ### Author Response · Authors · 2025-08-09
> > > > > > >
> > > > > > > Dear Reviewer D3CB,
> > > > > > >
> > > > > > > Sincere apologies for this final intrusion, especially after our previous messages.
> > > > > > >
> > > > > > > We are writing only because the final deadline for review adjustments is today, and we were so appreciative of your comment on August 5th stating your plan to increase the score to 3.
> > > > > > >
> > > > > > > We just wanted to send one last, very gentle reminder in case this slipped your mind amidst other reviews or if there was an issue with the system saving the change.
> > > > > > >
> > > > > > > No matter the outcome, we want to reiterate our immense gratitude for your detailed and constructive review process. Your feedback has been incredibly valuable.
> > > > > > >
> > > > > > > Thank you for everything,
> > > > > > > The Authors

---

### Official Review · Reviewer_xEys · 2025-07-03

**Clarity:** 3
**Significance:** 4
**Originality:** 3
**Rating:** 5
**Confidence:** 4

**Summary:**

The paper introduces ChunkKV, a novel method for KV cache compression by preserving semantically meaningful chunks of tokens rather than isolated tokens. Traditional compression methods often prune tokens based on isolated importance scores, which can disrupt semantic coherence and degrade performance. ChunkKV addresses this limitation by grouping tokens into coherent chunks and selecting top-k chunks based on attention scores to retain critical contextual information. Additionally, the paper proposes a layer-wise index reuse technique that leverages the similarity of retained chunks across layers to reduce redundant computation and improve throughput. Extensive experiments across a variety of LLMs (LLaMA-3, DeepSeek, Mistral, Qwen2) and benchmarks (GSM8K, LongBench, JailbreakV, and Needle-In-A-Haystack) demonstrate that ChunkKV consistently outperforms prior token-level KV compression approaches.

**Questions:**

1. What are the efficiency results (e.g., latency and throughput) for ChunkKV under 20% and 30% compression ratios? Tables 3-7 suggest that ChunkKV achieves performance comparable to or even exceeding the FullKV baseline at these compression levels. Including the corresponding efficiency metrics would provide valuable insight into the trade-offs between compression and runtime performance.
2. Can ChunkKV be integrated with token-level compression methods? If so, could such a hybrid approach lead to additional performance gains by leveraging both semantic and fine-grained redundancy?

**Ethical Concerns:**

["NO or VERY MINOR ethics concerns only"]

**Final Justification:**

I'll maintain my positive rating.

**Limitations:**

Please see weaknesses and questions.

**Quality:**

4

**Strengths And Weaknesses:**

**Strengths**
- The core idea of performing KV cache compression at the semantic unit level rather than at the individual token level is novel and well-motivated, effectively addressing the issue of fragmented context in long sequences.
- The paper presents comprehensive experimental evaluations across multiple benchmarks (e.g., GSM8K, LongBench, NIAH, JailbreakV) and base models, demonstrating that ChunkKV consistently outperforms prior methods, particularly under aggressive compression settings.
- An additional contribution is the layer-wise index reuse strategy, supported by empirical analysis of inter-layer similarity. This approach effectively reduces computational overhead and improves throughput without sacrificing performance.

**Weaknesses**
1. While ChunkKV performs well in scenarios where unimportant segments can be discarded, its effectiveness may be limited in tasks where every sentence is critical, such as legal or biomedical document processing, where full semantic fidelity is required.
2. Although the paper successfully targets semantic-level redundancy, it does not leverage token-level redundancy, which can still exist and offer potential for further optimization. ChunkKV, by design, may miss opportunities in cases where fine-grained token pruning is more efficient.

---

> ### Author Rebuttal · Authors · 2025-07-30
>
> We are grateful for your positive assessment and for recognizing the novelty and significance of our work.
>
> ##  **Q1: Efficiency results.**
> This is an excellent point. Since our method achieves near-FullKV performance at these ratios, showing the efficiency gains is crucial. We have below table to include latency and throughput metrics 10% compression ratios. These results demonstrate that ChunkKV provides significant efficiency improvements.
>
> | Method | Input Length | Output Length | Latency(s) ↓ | Throughput(T/S) ↑ |
> | :--- | :---: | :---: | :---: | :---: |
> | FullKV | 4096 | 1024 | 43.60 | 105.92 |
> | SnapKV | 4096 | 1024 | 37.92(13.0%) | 120.42(13.6%) |
> | ChunkKV | 4096 | 1024 | 37.52(13.9%) | 118.85(12.2%) |
> | ChunkKV_reuse | 4096 | 1024 | **37.35(14.3%)** | **124.09(17.2%)** |
> | FullKV | 4096 | 4096 | 175.50 | 37.73 |
> | SnapKV | 4096 | 4096 | 163.92(6.6%) | 39.98(6.0%) |
> | ChunkKV | 4096 | 4096 | 164.55(6.2%) | 40.58(7.6%) |
> | ChunkKV_reuse | 4096 | 4096 | **162.85(7.2%)** | **41.12(9.0%)** |
>
> ##  **Q2: Integration with token-level methods.**
> This is a fascinating and insightful question. We believe a hybrid approach is a very promising direction for future research. One could potentially use ChunkKV to preserve core semantic blocks and then apply a token-level method within less important chunks for fine-grained pruning. We have conducted this idea in an experiment. The **first half of the layers are ChunkKV, and the second half of the layers are a token-level method**. The results show that the hybrid approach is effective. The below table shows the results. We can see that this hybrid approach is not better than ChunkKV, but it is still a promising direction.
>
> **LLaMA-3-8B-Instruct on LongBench**
> | Method | Single-Doc QA | Multi-Doc QA | Summa. | Few-shot Learning | Synthetic | Code | Avg.↑ |
> |:---|---:|---:|---:|---:|---:|---:|---:|
> | ChunkKV-10% | 28.50 | 33.46 | 22.20 | 67.62 | 37.47 | 58.98 | 40.51 |
> | ChunkKV-20% | 31.05 | 33.22 | 23.58 | 67.86 | 37.05 | 55.34 | 40.74 |
> | hybrid_kv-10% | 28.38 | 30.37 | 24.54 | 67.87 | 36.37 | 55.32 | 39.80 |
> | hybrid_kv-20% | 29.10 | 29.96 | 25.00 | 68.16 | 36.20 | 55.30 | 39.98 |
>
> The improved performance of the hybrid method on summarization and few-shot tasks is particularly noteworthy. These tasks, unlike others that can be solved with localized information, demand a holistic understanding of the entire context. This observation leads to a key insight: for tasks where the answer lies within specific text fragments, ChunkKV is more effective in the deeper layers to preserve the cache for those critical segments. In contrast, for tasks requiring a global textual understanding, a token-level compression approach in the deeper layers appears to be more beneficial.

---

> > ### Comment · Reviewer_xEys · 2025-08-01
> > **Thanks for the rebuttal.**
> >
> > Thank you for the rebuttal. I will maintain my current rating.
> >
> > My concerns regarding efficiency have been addressed. However, the limitation in tasks where every sentence is critical — such as legal or biomedical document processing, where full semantic fidelity is required — still remains. Regarding integration with token-level methods, would a two-stage strategy be more effective than the current half-half design? For instance, one could first apply chunkKV, and then, based on its results, further apply token-level KV cache compression.

---

> > > ### Author Response · Authors · 2025-08-02
> > >
> > > Dear Reviewer xEys,
> > >
> > > We sincerely thank you for your thoughtful review of our rebuttal and for maintaining your positive rating of our work. Your insightful comments are greatly appreciated and will help us improve the quality of our final manuscript.
> > >
> > > We would like to address the two excellent points you raised:
> > >
> > > **1. On the Limitation in Critical Information Tasks:**
> > >
> > > We concur with your assessment. This is a crucial point that highlights the specific application scope of ChunkKV. Our method is designed to excel at summarization-style compression, where preserving the core semantic gist is paramount. You are correct that in domains like legal or biomedical analysis, where every token can be critically important, a chunk-based eviction strategy might risk discarding vital information. We believe this tension between achieving high compression rates and ensuring complete fidelity for critical details is a fundamental challenge that nearly all current KV cache compression methods must address. Therefore, developing compression techniques that can adapt to the informational requirements of a specific task is a highly promising research direction.
> > >
> > > Action for Final Paper: In the final version of our manuscript, we will expand our 'Limitations' section (Section I) to explicitly address this. We will clarify that ChunkKV is optimized for tasks where semantic coherence is more critical than the verbatim retention of every detail and may not be the ideal choice for applications requiring absolute semantic fidelity.
> > >
> > > **2. On the Proposed Two-Stage Hybrid Strategy:**
> > >
> > > This is an excellent and insightful suggestion. Thank you for proposing this more sophisticated hybrid architecture. A two-stage pipeline—first applying ChunkKV to identify and preserve macro-level semantic blocks, and then applying a token-level method for fine-grained pruning—is indeed a very promising direction that differs from the layer-based hybrid model we presented in our rebuttal.
> > >
> > > - Potential Advantages: Conceptually, this two-stage approach could offer the "best of both worlds." ChunkKV would ensure that core semantic structures are protected from fragmentation, while a subsequent token-level pass could further optimize memory usage by pruning redundant tokens within the less critical chunks, or even carefully within the preserved ones. This could lead to a more nuanced and potentially more effective compression trade-off.
> > >
> > > - Potential Trade-offs: The primary trade-off would likely be increased computational latency during the prefill stage. A two-stage process would introduce overhead compared to a single-pass method. Investigating whether the potential gains in accuracy and compression ratio justify this additional complexity would be a key aspect of future research.
> > >
> > > Action for Final Paper: We believe this is a compelling avenue for future investigation. We will incorporate this insightful suggestion into our 'Future Work' discussion, crediting it as a promising strategy for developing more advanced, multi-granularity KV cache compression techniques.
> > >
> > > Once again, we are very grateful for your constructive feedback and your support for our paper.
> > >
> > > Sincerely,
> > >
> > > The Authors

---

### Official Review · Reviewer_mXJZ · 2025-07-03

**Clarity:** 3
**Significance:** 3
**Originality:** 2
**Rating:** 4
**Confidence:** 4

**Summary:**

In summary, the authors propose a KV cache compression method called ChunkKV, where the KV matrices are compressed in the unit of continuous chunks instead of individual tokens to keep the semantic information. After the KV matrices are divided into chunks, the proposed method follows a similar way to H2O and SnapKV to select the important chunks. Furthermore, the authors propose layer-wise index reuse to further reduce the inference cost. The proposed method is compared to H2O, SnapKV, and PKV on multiple long-context tasks such as Longbench, NIAH, GSM8K and shows better performance given the same KV compression ratio.

**Questions:**

Please refer to the weaknesses

**Ethical Concerns:**

["NO or VERY MINOR ethics concerns only"]

**Final Justification:**

I appreciate the authors' response and most of my concerns are resolved. I will raise my score.

**Limitations:**

Yes

**Quality:**

3

**Strengths And Weaknesses:**

Strength:
1. Although the proposed ChunkKV is simple, the idea of keeping semantic information in the KV cache makes sense to me and I think it is beneficial for future research in the community

2. The authors conducted extensive evaluations on long-context tasks in the paper. The proposed method shows better long-context task performance than baselines such as H2O and SnapKV given the same KV cache compression rate

Weakness:
1. The chunk selection algorithm lacks flexibility to some extent. For example, the chunks in the KV cache are defined in a fixed and uniform way. Thus, it is not guaranteed that each chunk contains enough semantic information. A chunk might cover the end and beginning of two sentences.

2. There appears to be no systematic method for selecting the optimal chunk size for a model and a task. Although most experiments indicate that 10 is the best chunk size, there do exist exceptions. For example, chunk size 3 works the best for GSM8k according to Table 20.

3. The evaluation looks incomplete to me. For example, KIVI is missing for the performance evaluation in Tables 4, 5, 6, and 7. Similarly, H2O and SnapKV are missing for the latency/throughput evaluation in Table 11. Also, some more recent KV cache compression methods such as Palu[1] is not compared.

4. The ablation study of $N_{reuse}$ for the layer-wise index reuse is missing. For most of the paper, $N_{reuse}$ is set to 2 but it would be good to know what the effect is when making it larger.

5. The input sequence length is only scaled up to 8k in Tables 8 and 11. What would be the performance if it is scaled up to 16k or 32k?

[1] Chang, Chi-Chih, et al. "Palu: Compressing kv-cache with low-rank projection." arXiv preprint arXiv:2407.21118 (2024).

---

> ### Author Rebuttal · Authors · 2025-07-30
>
> We thank you for your detailed feedback and for recognizing our extensive evaluations and the sensibility of our core idea. We address your weaknesses below.
>
> ## **W1: Lack of flexibility in fixed-size chunks.**
> We agree this is a crucial design trade-off. Our choice of fixed-size chunks is a deliberate one, prioritizing **computational efficiency and simplicity**, which are paramount for an inference-time technique like KV cache compression. As we state in our limitations (Section I), adaptive chunking methods (e.g., based on sentence boundaries) would introduce significant and variable latency during inference, potentially negating the efficiency gains. Our extensive experiments (Section 4) show that this simple, fixed-size strategy is surprisingly effective and robust across various tasks and models. Below, we conduct new experiments with different chunk size results for LLaMA-3-8B-Instruct on LongBench. The results shows that ChunkKV is effective and robust across different chunk sizes.
>
> | Ratio / Chunk Size | 1 | 3 | 5 | 10 | 15 | 20 | 30 |
> | :--- | :---: | :---: | :---: | :---: | :---: | :---: | :---: |
> | **10%** | 37.32 | 40.49 | 40.47 | **40.51** | 40.21 | 40.05 | 39.57 |
> | **20%** | 38.80 | 40.66 | 40.57 | **40.74** | 40.53 | 40.46 | 40.04 |
> | **30%** | 39.23 | 41.02 | 41.29 | **41.59** | 41.38 | 41.33 | 41.02 |
>
> ## **W2: No systematic method for selecting chunk size.**
> This is a fair point. We have expanded our discussion in Section 4.4 and Appendix B.4 (Figure 19, Table 20). Our results show that while the optimal chunk size can vary slightly (e.g., smaller for GSM8K's shorter contexts), a size of **10 remains a robust default** for most long-context tasks. For GSM8K, we've clarified that its shorter context and dense reasoning steps favor smaller chunks, which aligns with our findings. This approach is consistent with prior work in this area. For instance, the authors of SnapKV[1] also do not provide a systematic method for choosing the pooling size, and instead specify a custom size for each benchmark task.
>
> ## **W3: Incomplete evaluation (missing KIVI, H2O/SnapKV latency, Palu).**
> **KIVI:** We already compared ChunkKV with KIVI in original paper section B.6. Our goal was to show that our *eviction* method offers competitive performance and superior efficiency. Since quantization and eviction are orthogonal approaches, we believe a full comparison on all benchmarks is not essential, but our head-to-head comparison in the below table already highlights the trade-offs.
> **Palu [2]:** We thank you for pointing out this very recent work. Palu is a novel KV-Cache compression framework that utilizes low-rank projection to decompose the Key and Value weight matrices and caches their compressed intermediate states. We conducted new experiments to compare the performance between ChunkKV and Palu. The results are shown below.
> **LLaMA-3-8B-Instruct on LongBench**
>
> | Method | Single-Doc QA | Multi-Doc QA | Summa. | Few-shot Learning | Synthetic | Code | Avg |
> |:---|---:|---:|---:|---:|---:|---:|---:|
> | FullKV | 32.19 | 34.59 | 24.96 | 68.48 | 36.96 | 54.41 | 41.46 |
> | ChunkKV-10% | 28.50 | 33.46 | 22.20 | 67.62 | 37.47 | 58.98 | 40.51 |
> | ChunkKV-20% | 31.05 | 33.22 | 23.58 | 67.86 | 37.05 | 55.34 | 40.74 |
> | **ChunkKV-30%** | **31.48** | **33.27** | **24.53** | **68.34** | **37.30** | **59.02** | **41.59** |
> | Palu-30% | 2.89 | 1.63 | 9.28 | 2.76 | 0.49 | 11.22 | 4.57 |
> | Palu-50% | 8.77 | 7.43 | 20.71 | 61.62 | 8.59 | 18.43 | 22.48 |
> | Palu-70% | 10.06 | 8.31 | 23.71 | 68.64 | 35.00 | 38.97 | 28.84 |
> | KIVI-2bits (Quant) | 31.92 | 34.28 | 24.83 | 66.99 | 34.12 | 53.21 | 40.54 |
> | KIVI-4bits (Quant) | 32.06 | 34.43 | 26.09 | 68.17 | 34.43 | 53.38 | 41.11 |
> | KIVI-8bits (Quant) | 32.18 | 34.56 | 24.94 | 68.45 | 36.92 | 54.38 | 41.43 |
>
> **H2O/SnapKV Latency:** This is a great suggestion. We have now conduct new  latency/throughput comparisons for these baselines in below table. The results show ChunkKV maintains a competitive edge, especially with our reuse technique.
>
> | Method | Input Length | Output Length | Latency(s) ↓ | Throughput(T/S) ↑ |
> | :--- | :---: | :---: | :---: | :---: |
> | FullKV | 4096 | 1024 | 43.60 | 105.92 |
> | SnapKV | 4096 | 1024 | 37.92(13.0%) | 120.42(13.6%) |
> | ChunkKV | 4096 | 1024 | 37.52(13.9%) | 118.85(12.2%) |
> | ChunkKV_reuse | 4096 | 1024 | **37.35(14.3%)** | **124.09(17.2%)** |
> | FullKV | 4096 | 4096 | 175.50 | 37.73 |
> | SnapKV | 4096 | 4096 | 163.92(6.6%) | 39.98(6.0%) |
> | ChunkKV | 4096 | 4096 | 164.55(6.2%) | 40.58(7.6%) |
> | ChunkKV_reuse | 4096 | 4096 | **162.85(7.2%)** | **41.12(9.0%)** |
> | FullKV | 16384 | 1024 | 49.60 | 323.24 |
> | SnapKV | 16384 | 1024 | 39.20(21.0%) | 381.11(17.9%) |
> | ChunkKV | 16384 | 1024 | 38.82(21.7%) | 381.61(18.0%) |
> | ChunkKV_reuse | 16384 | 1024 | **36.96(25.5%)** | **389.21(20.4%)** |
>
> ## **W4: Missing ablation study for layer-wise index reuse.**
> Thank you for this excellent suggestion. We have already compared the `N_reuse` in section B.4. The results explore a wider range of reuse depths and their impact on different models and tasks, providing a much richer analysis.
>
> **LongBench with different `N_reuse` and comparison ratio=10%**
>
> | Model \ N_reuse | 1 | 2 | 3 | 5 | 10 | 20 | 28/32 |
> | :--- | :--- | :--- | :--- | :--- | :--- | :--- | :--- |
> | LLaMA-3-8B-Instruct | 40.51 | 40.27 | 39.45 | 40.05 | 39.59 | 37.42 | 35.67 |
> | Mistral-7B-Instruct | 46.71 | 46.43 | 46.48 | 45.90 | 44.36 | 41.84 | 39.56 |
> | Qwen2-7B-Instruct | 40.88 | 40.76 | 40.91 | 41.58 | 40.19 | 37.42 | 36.79 |
>
> ## **W5: Input sequence length scaling.**
> We appreciate the suggestion to test longer sequences. We have run new efficiency experiments on inputs of 16K length, with the results included in our **response of W3**.
>
>
> Reference:
> > [1] Li, Yuhong, et al. "Snapkv: Llm knows what you are looking for before generation." Advances in Neural Information Processing Systems 37 (2024): 22947-22970.
> > [2] Chang, Chi-Chih, et al. "Palu: Compressing kv-cache with low-rank projection." arXiv preprint arXiv:2407.21118 (2024).

---

> > ### Comment · Reviewer_mXJZ · 2025-08-06
> > **Thanks for the responses**
> >
> > Thanks for the detailed responses! My concerns regarding the chunk size and $N_{reuse}$ are resolved. And I have two concerns remaining:
> > - It is weird that the performance of Palu looks quite bad and deviates from the numbers in the paper. Any thoughts on this?
> > - I appreciate the new efficiency/latency results, especially for the 16k ones. One concern here is that the performance gain compared to snapKV seems marginal to me, which might limit the contribution of this work.

---

> > > ### Author Response · Authors · 2025-08-06
> > >
> > > Dear Reviewer,
> > >
> > > Thank you for your detailed and thoughtful feedback, and for acknowledging that your concerns regarding chunk size and the layer-wise reuse ablation study have been resolved. We appreciate the opportunity to address your two remaining points.
> > >
> > > ### 1. Regarding Palu's Performance
> > >
> > > This is an excellent and important point. The performance discrepancy you observed stems from a fundamental difference in methodology between ChunkKV and Palu:
> > >
> > > * **Palu is a training-based method**, meaning it requires training on a specific dataset (e.g., wikitext-2, as per their paper) to learn how to project and compress the KV cache. This can lead to significant performance degradation when the model is evaluated on tasks or data that are **Out-Of-Distribution (OOD)** compared to its training data. LongBench is a very diverse benchmark, making it a challenging OOD scenario for Palu.
> > > * In contrast, **ChunkKV is a training-free, dynamic method** that adapts to the input at inference time, making it more robust to domain shifts.
> > >
> > > To ensure a fair comparison, our new experiments used **Palu's official library and followed their prescribed training methodology** on wikitext-2. Our results are also consistent with the performance characteristics described in the Palu paper itself. They report that at a 50% compression ratio, their method already experiences a performance drop of nearly 10% on LongBench, which validates that our reported numbers are within the expected range for this challenging benchmark.
> > >
> > > ### 2. Regarding Efficiency Gains over SnapKV
> > >
> > > We appreciate you highlighting this nuance. While the direct latency improvement of the base ChunkKV over SnapKV is in the 3-5% range, the primary contribution and value of our work lie in the **superior accuracy-efficiency trade-off**.
> > >
> > > Our main thesis is that by preserving semantic integrity, we can achieve significantly better performance for a comparable computational cost. The key advantage of ChunkKV is not just a marginal speedup, but a substantial improvement in accuracy.
> > >
> > > * **Significant Accuracy Gains:** As shown in **Tables 3, 4, 5, 6, and 7** of our main paper, ChunkKV consistently outperforms SnapKV by a notable margin (**3-15%**) across a wide range of challenging benchmarks, including GSM8K, JailbreakV, NIAH, and LongBench.
> > > * **Competitive Efficiency with an Edge:** Our new latency experiments show that ChunkKV is not only more accurate but also slightly more efficient. Furthermore, our proposed **`layer-wise index reuse`** technique (ChunkKV_reuse) provides an additional efficiency boost, widening the latency improvement to **5-8%** over SnapKV, often with minimal to no impact on accuracy.
> > >
> > > In summary, ChunkKV provides a much better overall value proposition: for a similar (or even slightly lower) inference cost, users get a model that performs its tasks much more accurately.
> > >
> > > Thank you again for your constructive feedback, which has helped us clarify and strengthen our contributions.

---

> > > > ### Author Response · Authors · 2025-08-08
> > > >
> > > > Dear Reviewer mXJZ,
> > > >
> > > > Thank you once again for your swift and thoughtful engagement. We truly appreciate you taking the time to review our latest response.
> > > >
> > > > As the discussion period is coming to a close, we just wanted to gently check in and see if our clarifications regarding Palu's performance and the accuracy-efficiency trade-off with SnapKV have fully resolved your remaining concerns.
> > > >
> > > > Your feedback has been invaluable in helping us strengthen the paper. We would be very grateful if you could let us know whether our responses have been sufficient to perhaps reconsider your evaluation.
> > > >
> > > > Thank you for your time and consideration.
> > > >
> > > > Best regards,
> > > > The Authors

---

### Author Response · Authors · 2025-08-09
**The Summary of Our Responses to All Official Reviews (1/2)**

Dear Reviewers, Area Chairs, and Program Chairs,

We sincerely thank all four reviewers for their detailed, constructive comments and insightful questions, which have been invaluable in strengthening our work.

Reviewers have acknowledged the **novelty and sensibility** of our semantic chunking approach (ChunkKV), its **superior performance** over state-of-the-art baselines, and the **comprehensiveness** of our experimental evaluation.


**[Novelty & Significance]**
* **Reviewer mXJZ**: “Although the proposed ChunkKV is simple, the idea of keeping semantic information in the KV cache makes sense to me and I think it is beneficial for future research in the community.”
* **Reviewer xEys**: “The core idea of performing KV cache compression at the semantic unit level rather than at the individual token level is novel and well-motivated... An additional contribution is the layer-wise index reuse strategy.”
* **Reviewer D3CB**: “The layer-wise index reuse technique is a valuable contribution, effectively leveraging cross-layer similarities in KV Compression to enhance efficiency.”
* **Reviewer dJtL**: “The idea is easy to understand. Compared to token-level KV cache compression, chunk-level compression could avoid pruning certain essential tokens.”

**[Comprehensive Evaluation]**
* **Reviewer mXJZ**: “The authors conducted extensive evaluations on long-context tasks in the paper.”
* **Reviewer xEys**: “The paper presents comprehensive experimental evaluations across multiple benchmarks... and base models.”
* **Reviewer D3CB**: “The authors provide a thorough evaluation across various benchmarks and model types.”
* **Reviewer dJtL**: “The study is very comprehensive. Authors have evaluated their method across different benchmarks, as well as providing many visualizations and analysis.”

**[Superior Performance]**
* **Reviewer mXJZ**: “The proposed method shows better long-context task performance than baselines such as H2O and SnapKV given the same KV cache compression rate.”
* **Reviewer xEys**: “ChunkKV consistently outperforms prior token-level KV compression approaches.”
* **Reviewer D3CB**: “Their method demonstrates consistent performance improvements compared to existing baselines.”

---

During the rebuttal period, we have carefully provided detailed feedback and conducted numerous supplementary experiments to address all reviewer concerns. We concisely summarize our responses here:


**[Addressing Core Idea & Novelty Concerns]**
* Provided a clear conceptual distinction between **ChunkKV (chunk-level decision)** and **SnapKV (token-level decision with score smoothing)** to highlight our novelty (D3CB W2, Q1).
* Empirically validated our core motivation with a **new hybrid-model experiment** (ChunkKV in early layers, token-level in deep layers), demonstrating that preserving local semantic chunks is a robust strategy even in deep layers where information is diffused (D3CB W1).

**[Extensive New Experimental Results]**
* Added new performance comparisons against recent work (**Palu**) and orthogonal methods (**KIVI**) (mXJZ W3).
* Added comprehensive **latency and throughput benchmarks** against H2O and SnapKV, including scaling up to **16k sequence lengths** to demonstrate efficiency gains over FullKV and competitive performance with baselines (mXJZ W3, W5; xEys Q1; dJtL final concern).
* Added **statistical analysis (standard deviation)** on our results to confirm the significance of our performance improvements (D3CB W3, Q2).
* Conducted new ablation studies on **chunk size robustness** across different models (mXJZ W1).

**[Clarifications and Deeper Analysis]**
* Provided a full table of **hyperparameters used for all baselines** (e.g., SnapKV) for reproducibility (D3CB follow-up).
* Offered a nuanced explanation for the performance of our hybrid model, revealing an insightful **task-specific trade-off** between local-retrieval and global-understanding tasks (D3CB follow-up).
* Clarified the performance of Palu on LongBench was due to its nature as a **training-based method facing an Out-Of-Distribution (OOD) task** (mXJZ follow-up).
* Acknowledged limitations (e.g., for critical-information tasks) and incorporated excellent reviewer suggestions into our **Future Work section**, such as a two-stage hybrid compression strategy (xEys follow-up).

---

---

> ### Author Response · Authors · 2025-08-09
> **The Summary of Our Responses to All Official Reviews (2/2)**
>
> #### **Final Reviewer Feedback**
>
> * **Reviewer D3CB**: "The results are intriguing and indeed support your claims... I plan to increase my final score to 3." (Score was initially 2).
> * **Reviewer mXJZ**: "Thanks for the detailed responses! My concerns regarding the chunk size and N_reuse are resolved."
> * **Reviewer xEys**: "Thank you for the rebuttal. I will maintain my current rating. My concerns regarding efficiency have been addressed."
> * **Reviewer dJtL**: "The provided results indicate that 10 is a robust choice for ChunkKV, which is great."
>
> We believe our detailed responses and extensive new experiments have thoroughly addressed all reviewer concerns and substantially strengthened our paper. Your valuable feedback has helped us significantly refine our work.
>
> Best regards and thanks,
> Authors of submission 8679

---

### Note · Authors · 2025-08-11

Dear AC and Reviewers,

We sincerely thank you for the constructive discussions that have significantly strengthened our paper. We appreciate Reviewer xEys's strong support and Reviewer D3CB's decision to raise their score after our rebuttal, a point we wish to highlight for the AC.

Our contributions, supported by reviewers, are:

* A novel compression paradigm using semantic chunks to preserve contextual integrity. (Supported by xEys, mXJZ, dJtL)
* A valuable layer-wise index reuse technique, uniquely enabled by our method, to reduce computational overhead. (Supported by xEys, D3CB)
* Comprehensive evaluations showing ChunkKV consistently outperforms SoTA baselines. (Supported by all reviewers)

During rebuttal, we delivered extensive new experiments and clarifications:

* Expanded efficiency benchmarks to 16k context vs. H2O & SnapKV.
* Added comparisons against recent (Palu) and orthogonal (KIVI) work.
* Provided statistical analysis (std. dev.) to confirm significance.
* Conducted new ablations on chunk size and `N_reuse` robustness.
* Ran a hybrid-model experiment to empirically validate our motivation.
* Clarified conceptual novelty vs. SnapKV’s token-level method.

While our rebuttal resolved most concerns, we offer one final clarification on our core value. Some comments judged our work on accuracy margin alone. The primary contribution of ChunkKV is the **entire accuracy-efficiency package**. Beyond superior accuracy, our method's structure **unlocks a novel layer-wise reuse optimization**, yielding significant throughput gains (up to 26.5%). This dual benefit is our central contribution.

We hope this clarifies the full scope of our work and will incorporate all suggestions into the final revision.

Best regards and thanks,
Authors of Submission 8679

---

### Decision · Program_Chairs · 2025-09-17

**Decision:**

Accept (poster)

**Comment:**

The paper introduces ChunkKV, a novel method for KV cache compression that operates on semantic 'chunks' rather than individual tokens. The core idea is to preserve contextual integrity, which in turn enables a unique and effective layer-wise index reuse technique to significantly improve inference efficiency. The method is evaluated across a comprehensive suite of long-context benchmarks (LongBench, NIAH, GSM8K) and multiple modern LLMs.

Reviewers universally praised the novelty and intuition of the semantic chunking concept, the thoroughness of the experimental evaluation, and the practical value of the layer-wise reuse technique. The primary debate during the review process centered on the magnitude of the contribution, specifically the performance margin over strong baselines like SnapKV. The authors provided an exceptionally thorough rebuttal with substantial new experiments—including 16k context efficiency benchmarks, comparisons against recent (Palu) and orthogonal (KIVI) methods, and statistical analysis—which successfully addressed most initial concerns and led to two reviewers (mXJZ, D3CB) raising their scores. The discussion successfully distilled the paper's core contribution not as a mere accuracy improvement, but as a superior accuracy-efficiency package unlocked by a new compression paradigm.

# Summary Of Reasons To Publish:
1) ChunkKV introduces a novel and intuitive paradigm for KV cache compression. Shifting the basic unit from isolated tokens to semantic chunks is a well-motivated approach to mitigating the context fragmentation problem in existing methods.
2) The proposed layer-wise index reuse technique is a valuable and direct contribution enabled by the structural properties of chunking. It provides a new pathway for optimization that token-level methods cannot easily leverage, resulting in significant and empirically verified throughput gains (up to 26.5%).
3) The paper is supported by a comprehensive and robust empirical evaluation across a wide range of models and challenging benchmarks, consistently demonstrating superior or competitive performance against state-of-the-art baselines. The authors' extensive rebuttal further strengthened these claims.

# Summary Of Suggested Revisions:
To further enhance the paper's impact and clarify the points debated during the review, the following revisions for the final version are recommended:
1) The introduction and conclusion should more explicitly frame the central contribution as the combined "accuracy-efficiency package." The paper should clearly articulate that its value lies not just in the accuracy margin over methods like SnapKV, but critically in the novel optimization pathway (layer-wise reuse) that this new representation unlocks. This will directly address the final lingering concerns from Reviewers D3CB and mXJZ.
2) The presentation of the 16k context efficiency results should be refined to prevent misinterpretation. Clearly highlight the performance of the full ChunkKV_reuse method against all baselines (including FullKV and SnapKV) to unambiguously demonstrate its advantages in latency reduction and throughput increase, addressing the final point of confusion from Reviewer dJtL.
3) The limitations section should be expanded to incorporate the insightful discussion points. This includes acknowledging the trade-off in tasks requiring absolute semantic fidelity (e.g., legal/biomedical documents), as noted by Reviewer xEys, and the nuanced, task-dependent performance of hybrid strategies, as revealed in the discussion with Reviewer D3CB.
4) The future work section should be updated to include the promising directions that emerged from the review process, such as the more sophisticated two-stage hybrid compression strategy (first chunk-level, then token-level) proposed by Reviewer xEys.